# Adaptive traits of cysts of the snow alga *Sanguina nivaloides* unveiled by 3D subcellular imaging

Jade A. Ezzedine [1,6], Clarisse Uwizeye [1,6], Grégory Si Larbi[1], Gaelle Villain [1], Mathilde Louwagie[1], Marion Schilling [1], Pascal Hagenmuller [2], Benoît Gallet [3], Adeline Stewart [1], Dimitris Petroutsos [1], Fabienne Devime[1], Pascal Salze[4], Lucie Liger [4], Juliette Jouhet [1], Marie Dumont [2], Stéphane Ravanel [1], Alberto Amato [1], Jean-Gabriel Valay[4], Pierre-Henri Jouneau [5], Denis Falconet[1] & Eric Maréchal [1] ✉

*Sanguina nivaloides* is the main alga forming red snowfields in high mountains and Polar Regions. It is non-cultivable. Analysis of environmental samples by X-ray tomography, focused-ion-beam scanning-electron-microscopy, physicochemical and physiological characterization reveal adaptive traits accounting for algal capacity to reside in snow. Cysts populate liquid water at the periphery of ice, are photosynthetically active, can survive for months, and are sensitive to freezing. They harbor a wrinkled plasma membrane expanding the interface with environment. Ionomic analysis supports a cell efflux of $K^+$, and assimilation of phosphorus. Glycerolipidomic analysis confirms a phosphate limitation. The chloroplast contains thylakoids oriented in all directions, fixes carbon in a central pyrenoid and produces starch in peripheral protuberances. Analysis of cells kept in the dark shows that starch is a short-term carbon storage. The biogenesis of cytosolic droplets shows that they are loaded with triacylglycerol and carotenoids for long-term carbon storage and protection against oxidative stress.

Red snowfields occur at high elevations in mountain ranges worldwide, and in sub-Arctic and sub-Antarctic regions, due to the proliferation of pigmented single-cell algae[1–3]. The development of red snow blooms in late spring was reported in the European Alps, from ancient times to the 18th century[4–6], and in the Arctic in the early 19th century[7]. The first microscopic observations of red snow samples showed high densities of pigmented cells[6,8], with an average diameter of 15.9 μm (from 8.5 to 31.7 μm)[8]. Initially believed to correspond to spores of a unicellular fungus[8], pigmented cells were subsequently confirmed to be microalgal cysts[9]. In spite of a multitude of studies in Alpine and sub-Polar regions, the taxonomic assessment of the main species forming snow blooms has remained elusive only until recently. Based on three DNA markers (18 S, ITS2, rbcL) combined with light and electron microscopy observations, blooms collected in various locations in Europe,

[1]Laboratoire de Physiologie Cellulaire et Végétale, Centre National de la Recherche Scientifique, Institut National de Recherche pour l'Agriculture, l'Alimentation et l'Environnement, Commissariat à l'Energie Atomique et aux Energies Alternatives, Université Grenoble Alpes; IRIG, CEA-Grenoble, 17 avenue des Martyrs, 38000 Grenoble, France. [2]Centre d'Etudes de la Neige, Université Grenoble Alpes, Université de Toulouse, Météo-France, CNRS, CNRM, 38000 Grenoble, France. [3]Institut de Biologie Structurale, Centre National de la Recherche Scientifique, Université Grenoble Alpes, Commissariat à l'Energie Atomique et aux Energies Alternatives; IRIG, 71 avenue des Martyrs, 38000 Grenoble, France. [4]Jardin du Lautaret, Université Grenoble-Alpes, Centre National de la Recherche Scientifique; 2233 rue de la piscine, Domaine Universitaire, 38610 Gières, France. [5]Laboratoire Modélisation et Exploration des Matériaux, Commissariat à l'Energie Atomique et aux Energies Alternatives, Université Grenoble Alpes; IRIG, CEA-Grenoble, 17 avenue des Martyrs, 38000 Grenoble, France. [6]These authors contributed equally: Jade A. Ezzedine, Clarisse Uwizeye. ✉e-mail: eric.marechal@cea.fr

North and South America, and Arctic and Antarctic regions were shown to be produced by a novel genus of Chlamydomonadales, named *Sanguina*, producing cysts enriched in carotenoids with a high level of astaxanthin[10].

Two *Sanguina* species have been identified so far[10–12]. *S. nivaloides* makes red cysts of 7.3–39.0 μm diameter and is found in red snow blooms worldwide; a second species named *S. aurantia* makes smaller orange cysts and is apparently less common. In mountain ranges of temperate regions, based on observations of snowfields[10] and the survey of soil environmental DNA along altitudinal gradients[13], *S. nivaloides* was shown to occur only above the timber line, where the snow cover remains during months. Since climate change reduces seasonal and perennial snowfields and accelerates glaciers' retreat worldwide[14–18], the question of a decline, and possible extinction of *Sanguina* species in many mountain ranges, is posed. Whereas cysts can overcome the stress of a long snow season, it is not clear whether *Sanguina* may be affected by the disappearance of the snow cover and be resilient enough to remain in a different climatic context. To address this question, data regarding the structural and functional bases of *S. nivaloides* adaptation to its natural habitats, including snowpack, is missing.

Until now, no cultivable strain of *S. nivaloides* has been obtained[10]. A recent study reports the successful culture of a green bi-ciliate cell of *S. aurantia*[11]. No transition from this green bi-ciliate form to a red cyst has been reported in laboratory conditions. The life cycle of *Sanguina* species is therefore still puzzling, without any conclusive evidence on the way they can actually colonize, proliferate, and populate the snowpack, although swimming vegetative cells are likely involved. We do not have any clue about the triggers and mechanisms of cyst formation. How cysts can convert back into vegetative or possibly sexual stages following snowmelt is also unresolved. Since we cannot control the growth and developmental stages of *Sanguina*, and cannot control any green-to-red cell conversion in laboratory conditions, our understanding of the adaptation of this species to the snow relies on the study of cells collected in their natural environment.

We analyzed cells in their in situ context, combining snow optical imaging, X-ray tomography, and physicochemical characterizations. We undertook an in-depth analysis of the three-dimensional subcellular architecture of *S. nivaloides* cysts, using focused-ion-beam scanning electron microscopy (FIB-SEM). This technique has been previously applied to chemically-fixed[19–22] and cryo-fixed microalgal cells[23,24], generating 3D reconstructions, suitable for quantitative morphometry (surface and volume) of organelles and subcellular structures. Based on these analyzes, we were able to highlight unique adaptive characteristics of *S. nivaloides* cysts that account for their long residence time in the snow environment.

## Results and discussion

### Detection of *Sanguina nivaloides* cysts in the liquid water fraction

Samples used for most analyzes were collected from a bloom in Vallon Roche Noir, at -2300 m a.s.l., in the French Alps (bloom 1; Fig. 1a, b) in 2021. Snow conductivity was $1.65 \pm 0.25$ μS/cm and pH $7.76 \pm 0.3$. The liquid water content (LWC), measured 10 cm below the snow surface, was $16 \pm 4$ mass%, reflecting the presence of liquid water, circulating mainly by infiltration from the melted surface[25]. We verified the presence of red cells using a field digital microscope: cysts were concentrated in the liquid water fraction (Fig. 1c). Although they lack flagella, the cysts moved along microscopic water currents at the edge of ice grains (Supplementary Movie 1). Low-energy X-ray tomography[26] of red snow carefully collected and conserved at −10 °C, at 10 μm resolution, allowed the visualization of ice grains and dust particles and of clusters of cysts (Fig. 1d) at the periphery of ice grains and at boundaries between grains that likely refroze during transportation from the field. No cyst was observed in the core of the ice grains.

A three-dimensional view of snow (Fig. 1e; Supplementary Movie 2) highlights algae clusters facing the reticulated air network, in which $CO_2$ circulates and can be captured via photosynthesis.

Three independent aliquot fractions were used for optical microscope observation one hour after sampling, confirming the presence of >95% red cysts in the >2 μm size fraction, not considering debris and dust. Cysts had $18.8 \pm 3.9$ μm diameter ($n = 200$ cells). They were kept at 4 °C in melted snow water (Fig. 1f) unless otherwise stated. No apparent alterations in morphology and size could be detected by routine optical microscopy observation after one-week storage in the dark at 4 °C.

Assessment of cell taxonomic identity was based on ITS2 sequence as described previously for *Sanguina* species[10]. We compared the ITS2 secondary structure of algal cells sampled from Vallon Roche Noire and all other blooms used in this study, to *S. nivaloides* holotype RS 0015-2010 and *S. aurantia* holotype RS 0017-2010. All sequences matched holotype RS 0015-2010 (Fig. 1g). Taxonomic assessment was further confirmed based on rbcL sequence phylogenetic reconstruction (Supplementary Fig. 1). Prominence of *S. nivaloides* was verified in all other blooms also analyzed in this work.

### Cyst photosynthesis is affected by freezing at −5 °C

Photosynthesis was evaluated on cysts stored either in melted snow at +4 °C and 0.1 μmol photons s⁻¹m⁻² or frozen 16 hours at −5 °C. In addition, a sample kept for over a year at +4 °C and 0.1 μmol photons s⁻¹m⁻² in a cold room was analyzed (Fig. 2).

Based on these results, cysts appear as photosynthetically active when freshly collected (Fig. 2a, b, g–j). At steady state photosynthesis lighting conditions, the effective photochemical quantum yield (Y(II)) indicates the amount of energy used in photochemistry in photosystem II, i.e. for the production of ATP and NADPH. Y(II) reflects therefore the capacity to assimilate $CO_2$. Its value is lower at high light (Fig. 2a, L2) compared to low light (L1), indicating the capacity of *S. nivaloides* to capture carbon under the high irradiance conditions that characterize the upper layer of snowpack. Consistently, the relative electron transfer rate (rETR) is higher at high light (Fig. 2i) compared to low light conditions (Fig. 2h). Under high light, non-photochemical quenching (NPQ) is activated (Fig. 2b, L2). The Fv/Fm value of $0.52 \pm 0.01$ measured after dark adaptation, reflecting the maximum efficiency of PSII, indicates that the algae are not in their best physiological condition, yet it is in the range of that measured commonly in green algae[27]. Low-temperature fluorescence emission spectra of freshly sampled *Sanguina nivaloides* highlight peaks at 685.4 and 713.5 nm corresponding to functional photosystem II and I, respectively (Fig. 2j). Following freezing at −5 °C, all photosynthetic parameters were affected (Fig. 2c, d, g–i), with a dramatic loss of NPQ (Fig. 2d), by contrast with other algal cells collected in the snow, such as the Trebouxiophyceae *Lobosphaera incisa*, which cells were reported to be resilient to freezing[28]. By contrast, after 12 months of conservation at +4 °C in the dark, photosynthetic parameters were still measurable and coherent with an active machinery (Fig. 2e, f, g–i), although showing a stronger sensitivity to high light conditions (Fig. 2i) and lowered capacity to activate NPQ mechanisms. This suggests that *S. nivaloides* cysts benefit from the thermal protection provided by the snow, and are able to survive for months and possibly years unless the temperature decreases strongly below the freezing point. In addition, lethality of *S. nivaloides* populations may therefore be very important after the snow melts, when cysts are at bare ground surface and exposed to late-season freezing episodes or icy winds.

### Association of *S. nivaloides* cysts with a community of bacteria

For electron microscopy analysis, the cells were either chemically fixed directly after collection (Supplementary Movies 3–4) or subjected to cryo-fixation after five days of storage in the dark at 4 °C (Supplementary Movies 5–6). Chemical treatment is practical for fieldwork, as

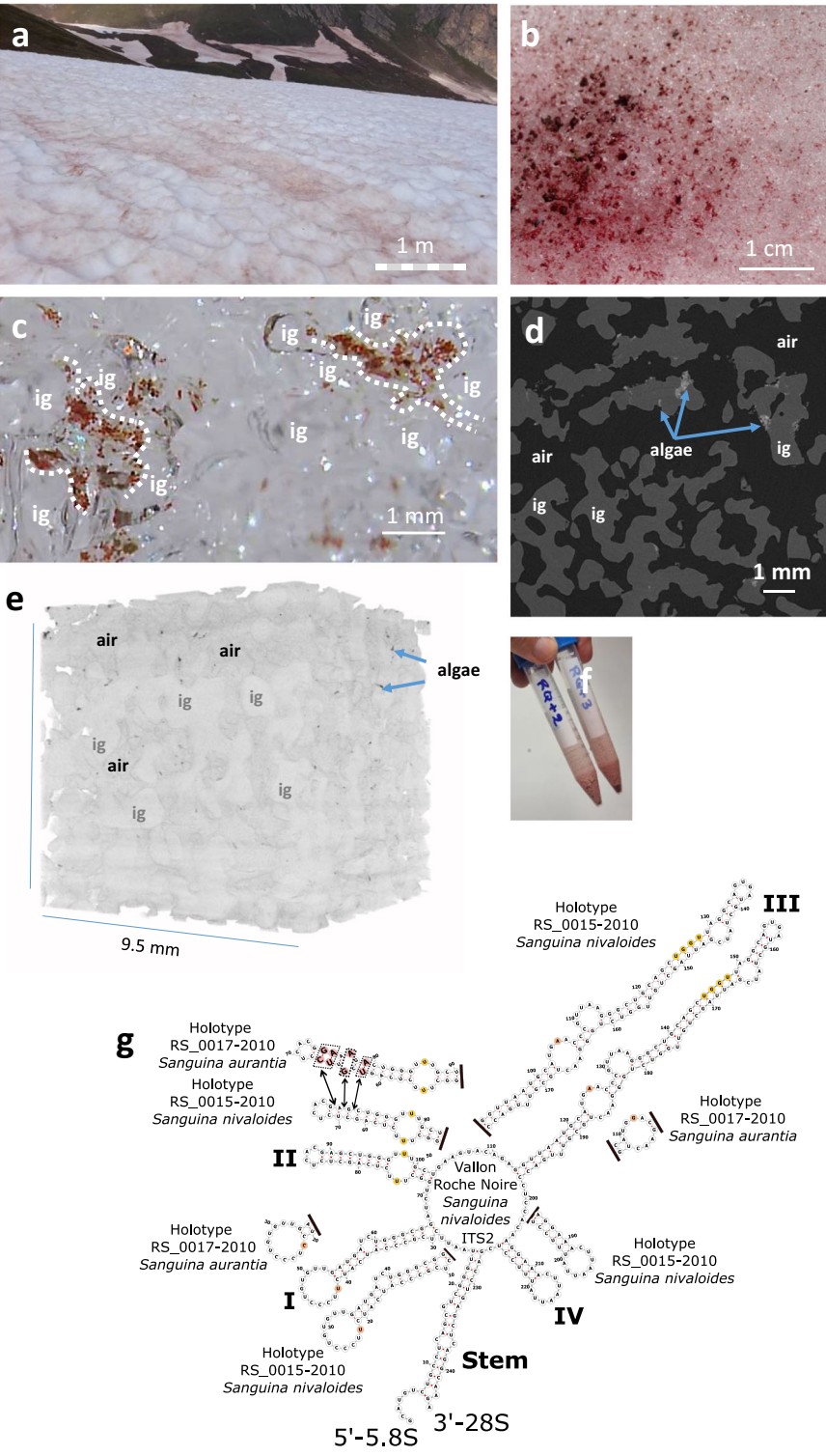

it allows a rapid fixation of cells in their native state. General ultrastructural features observed using both chemical and cryo-fixation methods can usually be compared[29], but cryo-fixation allows better preservation of macromolecular structures[30]. Here, cryo-fixation performed better in preserving lipid droplet (LD) structure at nanoscale level. The ultrastructure analysis of multiple cells, did not show any striking variation in the general organization of *S. nivaloides* cysts.

The red snow samples contained mainly algal cysts, mixed with dust and debris of various origins, and bacteria, with a ratio of >10 bacterial cells per cyst (Fig. 3a–c; Supplementary Movies 3–6). The cell wall of *S. nivaloides* cysts had a smooth surface, as reported

previously[10,12,31]. Its thickness ranged from 66 to 154 nm, as evaluated on 12 individual cells observed by FIB-SEM, in line with previous observations[10,12].

The majority of observed bacteria formed polysaccharide capsules, enclosing multiple cells. They could be found at the vicinity of cysts without any contact site or any visible hybrid structure involving *S. nivaloides* cell wall components (Fig. 3d, e; Supplementary Movie 7). In the microbiota from red snowfields from Finland, Norway, Sweden, and USA[32] or Japan[33], Proteobacteria were dominant. Snow Proteobacteria were further reported to develop some beneficial interactions with cultivable microalgal species[34]. We did not observe any fungal

**Fig. 1 | *Sanguina nivaloides* blooms. a** Red snowfield in Vallon Roche Noire, 2300 m. a.s.l. The bar shows the scale in front of the perspective view. **b** Algal bloom view at the melting snow surface. **c** Red cysts in the liquid water fraction circulating between ice grains. The liquid water content, measured 10 cm below the snow surface, was 16 ± 4 mass %. Imaging with field digital microscope shows cysts only in the liquid water fraction, moving along water currents in interstices between ice grains. This observation was repeated 5 times with similar result. **d** X-Ray imaging of red snow. Air is shown in black, ice grains in dark gray, dust particles in white, and clusters of cysts in light grey (arrows) at the periphery of ice grains. **e** Volume view of red snow analyzed by X-Ray tomography. Cysts are detected at the periphery of ice grains, facing the reticulated air network. **f** Collected algal cells for laboratory analysis. Although present at the surface of snow, cysts sedimented after melting.

**g** Assessment of algal species present in collected blooms based on ITS2 analysis. ITS2 secondary structures (including the 'stem' 5'− 5.8 S rRNA and 3'− 28 S rRNA) of algae sampled in Vallon Roche Noire and all other locations in this study, were predicted using RNAfold according to centroid algorithm. Obtained ITS2 structures were compared to *S. nivaloides* holotype RS 0015−2010 and *S. aurantia* holotype RS 0017−2010. All sequences matched holotype RS 0015-2010. Differences with holotype RS 0017−2010 are highlighted in pink, where additional complementary bases are present in helix II (5'-CG-3', 5'-UA-3', 5'-GA-3'). Single bases colored in brown are also different in helix I and III of *S. aurantia* in comparison to *S. nivaloides*. Pyrimidine - pyrimidine mismatch in Helix II and (G/U)GGU motif in helix III, specific of Viridiplantae, are shown in yellow. Black lines indicate no difference downstream of the structure. ig, ice grain.

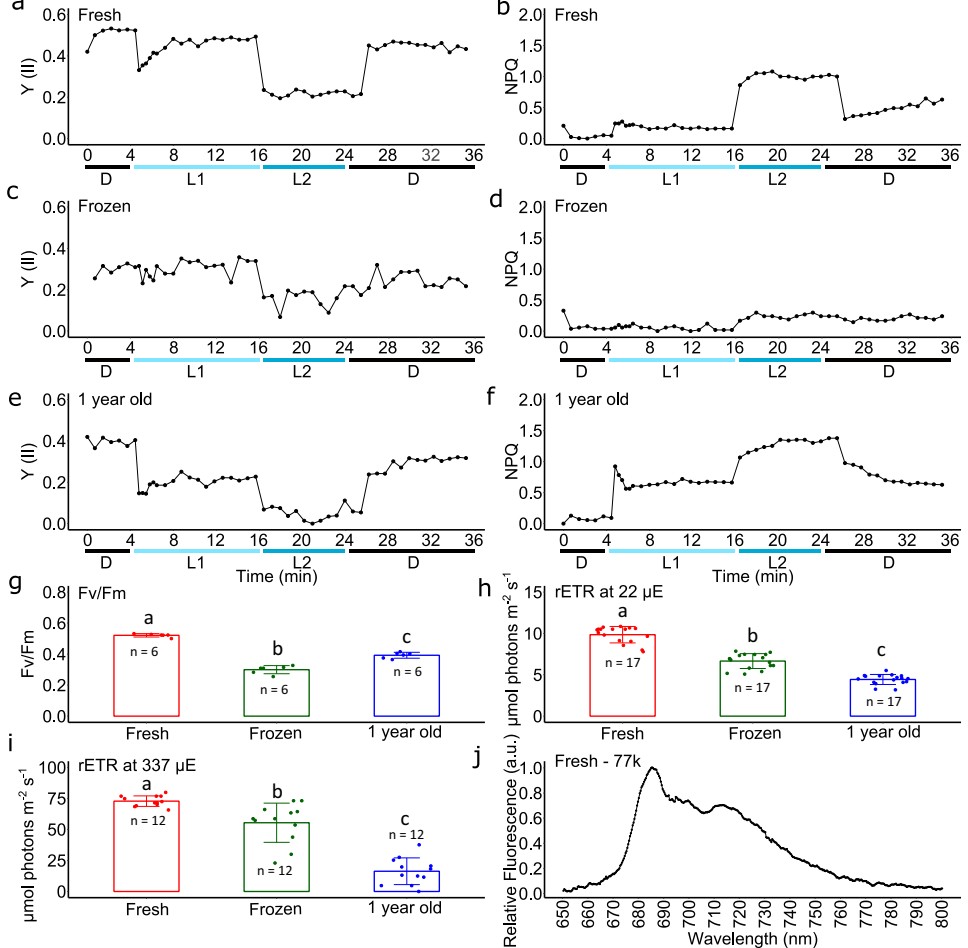

**Fig. 2 | Analysis of photosynthesis of *Sanguina nivaloides* collected within red snowfields in Vallon Roche Noire after various periods of conservation.**
**a, b** Effective photochemical quantum yield of photosystem II (Y(II)) and non-photochemical quenching (NPQ) in freshly-collected cells. **c, d** Y(II) and NPQ, in cells frozen 16 hours at -5 °C. **e, f** Y(II) and NPQ, in cells kept one year at + 4 °C. Prior to the onset of the measurements, cells were acclimated to darkness for 15 min. Chlorophyll fluorescence was recorded under different intensities of actinic light; starting with measurements in the dark (D), then at 22 (L1) and 337 (L2) μmol photons m⁻² s⁻¹ followed by 10 min of relaxation in the dark (D). **g−i** Fv/Fm, and relative photosynthetic electron transfer, rETR, at 22 and 337 μE of freshly collected samples, frozen samples, and samples stored one year at + 4 °C. Fv/Fm, rETR 22 and rETR 337 mean values ± SD were based on *n* = 6, 17 and 12 independent

measurements, respectively. For "fresh *vs*. frozen" comparisons, the *P-value* for Fv/Fm measurements based on a One-Way ANOVA test was 2.2 ×10⁻¹¹. For rETR 22 and rETR, 337 measurements, *P-values* based on a Kruskal-Wallis test were 3.0 × 10⁻¹⁰ and 1.2 × 10⁻⁶, respectively. One-sided ad hoc tests confirmed significant differences, shown with "a", "b" and "c" letters on the barplots. *P-values* from Tukey HSD test for Fv/Fm for the samples "fresh *vs*. frozen", "fresh *vs*. 1 year old" and "frozen *vs*. 1 year old" were 0.1 × 10⁻⁸, 0.1 × 10⁻⁸, and 2.0 × 10⁻⁶, respectively. *P-values* from Dunn's test with Boneferroni correction for ETR 22 were 1.2 × 10⁻³, 0.1 × 10⁻⁵, and 1.6 × 10⁻³, respectively. *P-values* for ETR 337 were 3.3 × 10⁻², 0.1 × 10⁻⁵, and 5.1 × 10⁻³, respectively. **j** Low temperature fluorescence emission spectrum of a fresh sample of *Sanguina nivaloides*. The obtained data were normalized to the photosystem II emission peak at 685 nm.

cells, possibly present in very low proportion, nor any pathogenic snow chytrids as reported recently in high latitude snow blooms[31]. Although bacteria were in the vicinity of *S. nivaloides* cell surface (Fig. 3d, blue arrows), suggesting that respective cell walls may interact

physically, we did not notice any direct contact site between algal and bacterial cell membranes. The absence of apparent physical cell-to-cell membrane contact sites suggests that biotic interactions may rely on the exchange of soluble material transferred via the plasma

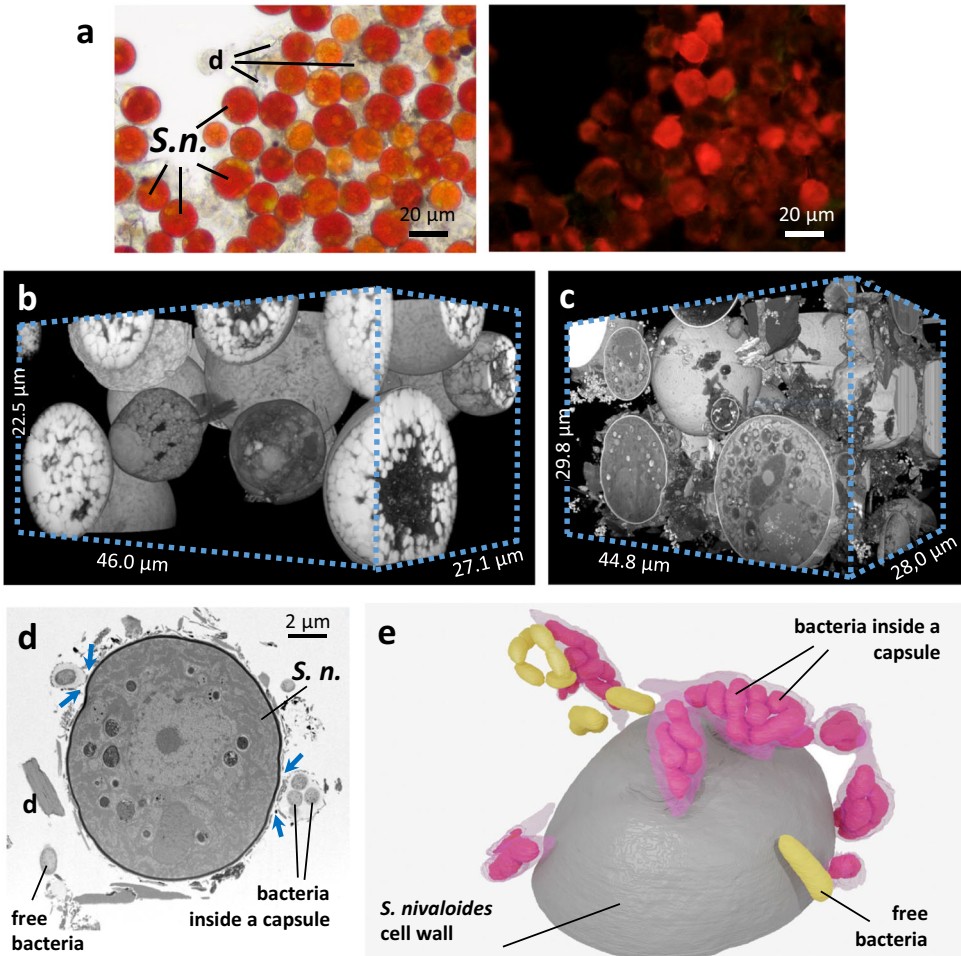

**Fig. 3 | FIB-SEM imaging of chemically-fixed and cryo-fixed *S. nivaloides* cysts.**
**a** Light microphotographs of red snow. Right, bright field image of mature cysts, with lipids droplet visible inside cells, pigmented in red or orange due to the presence of astaxanthin and other carotenoids. Left, chlorophyll autofluorescence (excitation 495 nm; emission 521 nm). Fluorescence intensity is higher in orange cells and cells with apparent green spots. Debris and dust in the sample have no pigmentation nor fluorescence. This observation has been repeated 6 times with similar results. **b** Volume view of chemically-fixed cells analyzed by FIB-SEM. One FIB-SEM analysis has been performed with this fixation method. **c** Volume view of cryo-fixed cells analyzed by FIB-SEM. One FIB-SEM analysis has been performed with this fixation method. **d** Detection of bacteria associated to *Sangina nivaloides* cysts. Image from a FIB-SEM stack showing free and encapsulated bacteria. This observation has been repeated with similar result on all images from the FIB-SEM stack. **e** Three-dimensional model of bacteria at the vicinity of a *Sangina nivaloides* cyst. S. n. *Sangina nivaloides*, d debris.

membranes of each partner, without any physical bridging or tethering systems.

## *S. nivaloides* cysts exhibit a wrinkled plasma membrane

The plasma membrane of cysts has numerous wrinkles at its surface (Fig. 4a). These wrinkles were homogenous in size, having a $406 \pm 90.9$ nm length and $140.6 \pm 44.2$ nm width. They do not intersect and do not follow any clear orientation, either parallel or in rosette shapes, covering the complete cell surface (Fig. 4b). When we compare the three-dimensional cell reconstruction of the wrinkled plasma membrane of a *S. nivaloides* cyst, having an area of $825.5 \, \mu m^2$, with the same surface without any wrinkle, having an area of $736 \, \mu m^2$, more than 12% of the surface is gained by the presence of the membrane wrinkles (Fig. 4b).

This wrinkled pattern likely reflects specific function of the cyst plasma membrane. A first hypothesis could be that the wrinkled plasma membrane would provide a three-dimensional matrix of molecular machineries involved in the biosynthesis of the cell wall. However, the cell wall per se appears as smooth and even. We did not detect any striking variations in its thickness, production of ornate features or peripheral mucus, or any sign of a remodeling or

development of cell-to-cell tethering structures (Figs. 3 and 4). The wrinkled pattern is therefore unlikely related to any structural pattern, roughness or physical reorganization of the cell wall.

Another possible role could be linked to the increase of the interface area with the environment for an intensified activity of specific transporters and receptors. Fluxes of chemicals are expected from the observed proximity with bacteria cells[34] (Fig. 3d, e), which suggests the existence of exchanges of organic molecules, nutrients and/or infochemicals. We wondered whether some important ionic exchanges with the liquid environment may also operate at the surface of cysts. We analyzed the elemental composition of snow by inductively coupled plasma mass spectrometry (ICP-MS), after a cautious removal of all microorganisms and solid debris by filtration (0.2 μm) (Fig. 5). This method did not allow the analysis of N or C, or such elements as Cl, nor of organic molecules. We analyzed the composition of red snow, and of white snow collected less than three meters from the bloom limits. Postulating that soluble elements were excluded from ice crystals, molar concentrations were calculated based on a LWC of 15%.

In all samples, the elemental concentrations were in the 5-50 μM range, for Na, Mg, K and Ca; 0.1–1.5 μM range, for Al, P, Mn, Fe, and Zn;

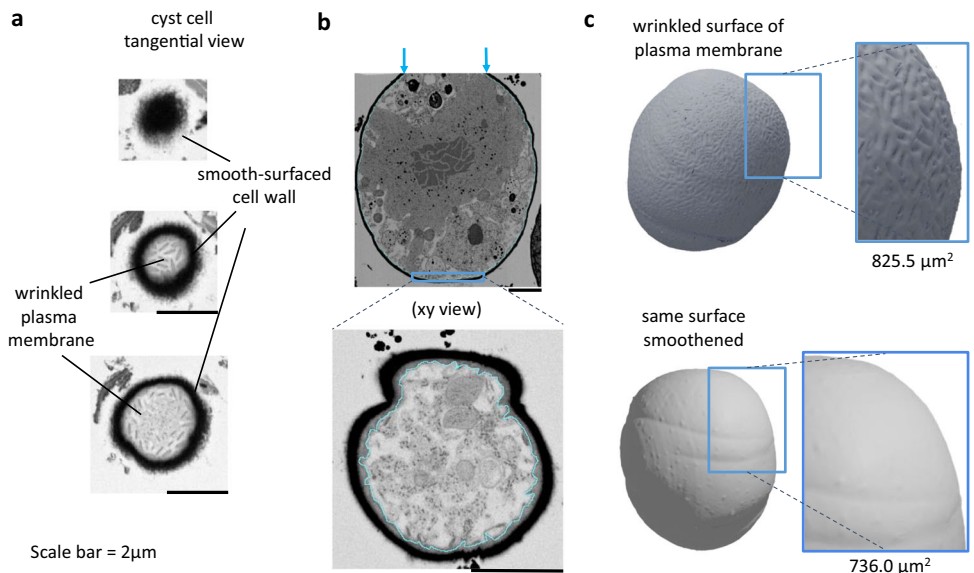

**Fig. 4 | Wrinkled plasma membrane of *Sanguina nivaloides* cysts. a** Tangential views of plasma membrane and cell wall in cross sections observed by FIB-SEM at variable depths**. b** Process of segmentation of plasma membrane based on EM image stacks. **c** Three-dimensional reconstruction of *Sanguina nivaloides* cyst plasma membrane. The 3D model with a wrinkled surface was filtered using the HC

Laplacian smoothing method[77]. The filter builds a new mesh based on the information of the average of the nearest vertices, and thus produces a smooth surface after three iterations. Based on the native wrinkled membrane and the computed smoothened one, surfaces are calculated and highlight a 12% area increase attributable to the plasma membrane architecture.

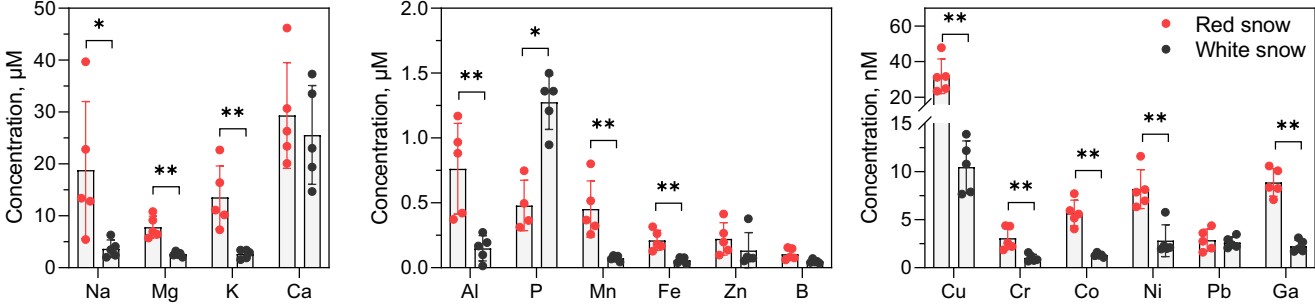

**Fig. 5 | Analysis of the soluble elements in the liquid water fraction of red and white snow.** Elements were analyzed by ICPMS, and concentrations were normalized based on a 15% LWC, and expressed in M. Histograms were split based on the elemental concentration ranges, from 5–50 μM for the most abundant to

<10 nM. Some elements such as C and N were not analyzed by this method. Analyzes was performed in 5 replicates. *P-value* was based on a Mann–Whitney test and considered significant as follows: *, *P*-value = $2.10^{-2}$; **, *P*-value = $8.10^{-3}$.

10–100 nM range for B and Cu, and lower 10 nM range, for Cr, Co, Ni, Pb, Ga (Fig. 5), reflecting the fact that the snow is an oligotrophic habitat. The composition of white snow is consistent with ions deriving from the alteration of minerals, such as feldspar[35], found abundantly in the Lautaret area. These elements could be detected in snow samples collected in Mont Brévent, France, 110 km distant in the same mountain range (Supplementary Fig. 2). They were also present in red and white snow samples collected in Mount Olympus, Greece, where the underlying rocks are predominantly calcareous, supporting a contribution from non-rock sources, such as directly from snowfall and dust aerosols (Supplementary Fig. 2). In all samples, the concentration of most elements was significantly higher in red compared to white snow (Fig. 5; Supplementary Fig. 2), suggesting either that blooms developed in areas enriched in nutrients or that, in the course of melting and bloom development, a broad process of ionic concentration occurred. We compared variations in the same concentration ranges and detected some distinctive features for some elements, which could be attributable to the presence of cysts. A first striking feature is the 5-25 fold increase in K concentration in red snow liquid water, which was always higher than that observed for other elements

in the same concentration range (Na, Mg), balanced by the relative stability of Ca level (Fig. 5; Supplementary Fig. 2). This suggests that *S. nivaloides* cysts and companion bacteria may release cations in the environment, possibly increasing, even moderately, the salinity of their own liquid environment. The light-dependent efflux of K through plasma membrane channels has been shown in some unicellular green algae[36]. A similar type of control of cyst plasma membrane ionic channels by light may explain this observation. Less abundant essential metals (Fe, Mn, Zn, Cu, Co, Ni) also displayed significant accumulation in red versus white liquid snow (Fig. 5; Supplementary Fig. 2). This trend probably illustrates a complex regulation of metal homeostasis in cysts, including their transport at the plasma membrane, to ensure the cellular demand is met[37]. The maintenance of photosynthesis in cysts (Fig. 2) illustrates the ability of *S. nivaloides* to maintain at least a supply of Fe, Mn, or Cu, which are essential to this process, despite low availability in the environment.

A second striking feature concerns P, which decreases sharply (Fig. 5) or increases less compared to other elements in the same concentration range (Supplementary Fig. 2). In conditions where P falls down to such a low level as 0.1 μM, the environment can be considered

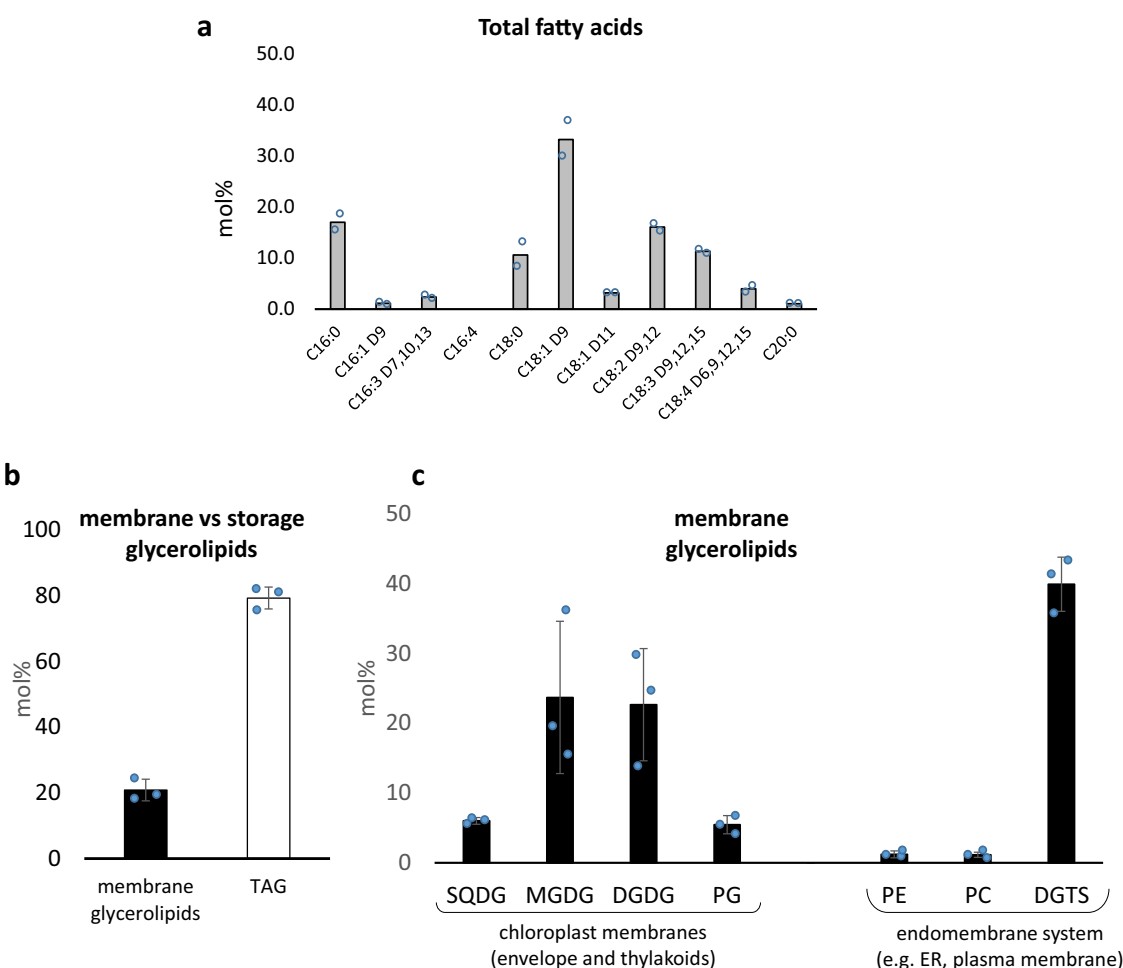

**Fig. 6 | Quantitative analysis of fatty acids and glycerolipids in *S. nivaloides* cysts. a** Fatty acid profile in a total lipid extract. Lipids were extracted from *S. nivaloides* cells collected from bloom 2, and subjected to a methanolysis. Obtained fatty acid methyl esters were identified by mass spectrometry and quantified by gas chromatography coupled to flame ionization detection. Fatty acids are represented based on their carbon-chain lengths (from 16 to 20 carbon), and the positions of double bonds (D*x*, numbered at the *x*th carbon from the carboxylic end). Relative proportions are expressed in mol%. The result is the average of two replicates. **b** Relative proportions of membrane and storage glycerolipids. Relative proportions are expressed in mol% of total glycerolipids. **c** Relative proportions of membrane glycerolipid classes. Relative proportions are expressed in mol% of membrane glycerolipids. Profiles shown in (**b**) and (**c**) were obtained from lipids extracted from bloom 4. Each result is the average of three replicates ± SD. DGDG digalactosyldiacylglycerol, DGTS diacylglyceryl-tri-methylhomoserine, MGDG monogalactosyldiacylglycerol, PC phosphatidylcholine, PE phosphatidylethanolamine, PG phosphatidylglycerol, SQDG sulfoquinovo-syldiacylglycerol, TAG triacylglycerol.

as P-limited, and the cell as P-starved. This further suggested an activity at the plasma membrane of phosphate transporters, and possibly other essential nutrients, such as N not analyzed here.

### Cyst lipidome confirms a lack of phosphate

We sought whether cysts presented physiological signatures of the environmental P depletion detected by ICP-MS. To that purpose, we analyzed the total profile of glycerolipids extracted from a bloom aliquot fraction, to detect whether any phosphorus-to-non-phosphorus lipid remodeling had occurred. A total lipid extract of bloom 2 ($1.5.10^6$ cells) was used to characterize all glycerolipids present. An aliquot fraction ($1.10^6$ cells) was first subjected to a methanolysis and obtained fatty acid methyl esters were identified and quantified by gas chromatography coupled to flame ionization detection (GC-FID) (Fig. 6a). Detected fatty acids are close to those found in extracts from *Chlamydomonas reinhardtii*[38,39] or *Chlorella* sp.[40], with fatty acids of 16 or 18 carbon chain lengths and up to 4 double bonds. We then separated the major glycerolipid classes by thin-layer chromatography, and identified three membrane plastid lipids, i.e. monogalactosyldiacylglycerol (MGDG), digalactosylglycerol (DGDG), sulfoquinovosyldiacylglycerol

(SQDG), two non-plastid lipids, i.e. phosphatidylcholine (PC) and the betaine lipid diacylglyceryl-trimethylhomoserine (DGTS), and the storage lipid triacylglycerol (TAG). PC and DGTS are common non-plastid lipids in chlorophyta, with the notable exception of *C. reinhardtii*, which does not synthesize PC[38,39]. In red snow extracts we did not exclude that PC might correspond to that in *S. nivaloides* and/or bacteria. Nevertheless, the very low level of phosphatidylethanolamine (PE) suggests that bacterial lipids are minor. Each membrane lipid harbors two acyl groups, which position on the glycerolipid structure (*sn*-1 and *sn*-2 positions of the glycerol backbone) was identified by ion trap MS (Supplementary Table 1), as described earlier[41]. The positional composition of acyls in TAG was also determined (Supplementary Table 2). We then quantified the relative proportion of each lipid class by liquid chromatography coupled to MS (LC-MS) in bloom 2 and bloom 4. In LC-MS analyzes, PE, and phosphatidylglycerol (PG) were also detected. Acyl profiles of all lipid classes were consistent between bloom 2 and 4 (Supplementary Fig. 3). Relative proportion of each lipid class (in mol%) was assessed as described previously[42].

There are three conserved lipidomic responses of photosynthetic organisms to phosphate starvation. Firstly, like in other nutrient

starved conditions, TAG accumulates[41,43,44]. Secondly, phosphatidylglycerol (PG) from photosynthetic membranes decreases and is replaced by SQDG[45–47]. Thirdly, non-plastidial phospholipids, usually PE and PC, are hydrolyzed and replaced by non-phosphorus lipids, such as DGTS or DGDG[45–48]. In total lipid extracts from *S. nivaloides* blooms, TAG represents 50-80% of total glycerolipids, plastidial PG level is always lower than that of SQDG and in non-platidial lipids, PC and PE are minor whereas DGTS represents 20-40% of total membrane glycerolipids (Fig. 6b; Supplementary Fig. 4). The glycerolipidome of *S. nivaloides* cysts provides, therefore, a physiological evidence that it has undergone an intense membrane lipid remodeling, with an accumulation of TAG and SQDG-to-PG and PC/PE-to-DGTS replacements, consistent with a response of the alga to a phosphate depleted environment (Fig. 5).

A striking feature of *S. nivaloides* cyst lipidome is its high level of polyunsaturated fatty acids (PUFA) in lipids making up photosynthetic membranes, i.e. MGDG, DGDG, and SQDG (Supplementary Fig. 3). In particular, these lipids contain 18:4 levels, unreported before in *C. reinhardtii* and *Chlorella* sp (Supplementary Table 2) in addition to 16:4, classically encountered in plastid lipids[49]. This high PUFA level is coherent with the maintenance of membrane fluidity in photosynthetic membranes at a low temperature[50–52] and a functional photosynthetic activity (Fig. 2). The detection of oxidized forms of MGDG (Supplementary Table 2), commonly encountered in photosynthetic cells, further suggests that cysts may be also exposed to some kind of oxidative stress, possibly related to high light exposure.

The high proportion of TAG, filling up the core of cytosolic LDs was expected given the number of LDs in the cytosol of *S. nivaloides*. TAG is a carbon storage form, providing fatty acids to mitochondrial or lysosomal beta-oxidation, a catabolic pathway supplying acetyl-CoA to the mitochondrial Krebs Cycle. Since two major forms of carbon reserves exist in green algae, i.e. TAG in the cytosol and starch in the stroma of chloroplast, we addressed whether some cytological features may provide evidence of their relative metabolic utilization in *S. nivaloides* cysts.

## Organization of organelles involved in energy and carbon metabolism

A recent comparative 3D morphometric analysis in representative microalgae genera (including Archaeplastida; i.e. *Micromonas* and *Galdieria*; Hacrobia, i.e. *Emiliana*; Alveolata, i.e. *Symbiodinium*; and Stramenopiles, i.e. *Phaeodactylum*, *Pelagomonas,* and *Nannochloropsis*), with cell sizes ranging from 2 to 200 μm³, highlighted a relatively constant occupancy by the main organelles, and preserved volumetric ratios between plastids and mitochondria[24]. In this quantitative survey, about half (40–55%) of the volume of an active planktonic cell is usually occupied by the nucleus (5–15%), plastids (15–40%) and mitochondria (2.5–5%), conservation of organellar volumes being a signature of evolutionary constraints preserving cellular functions (gene expression, energy production, and consumption, compartmentation of metabolic pathways)[24]. In addition, a correlation was shown between plastids and mitochondria volume ratio (~8) and surface area ratio (~2), together with the existence of contact sites, reflecting an inter-organelle relationship relevant for carbon assimilation and consumption to fuel vital cell processes[24].

We performed 3D morphometric analyzes on *S. nivaloides* cysts chemically or cryo-fixed (Fig. 7a, cells 1 and 2, respectively). The general 3D morphometric parameters are different from those classically observed for single-cell algae, with a nucleus taking a much smaller place than usual (2.3-3.6%), and volumes occupied by mitochondria (1.8-2.2%) and plastids (28.6-31%) in the lower range of subcellular occupancy usually reported for these organelles (Fig. 7a). This is evidently attributable to the abundance of lipid droplets filled with astaxanthin, representing 13.5-35.5% of the cell volume (Fig. 7a, b).

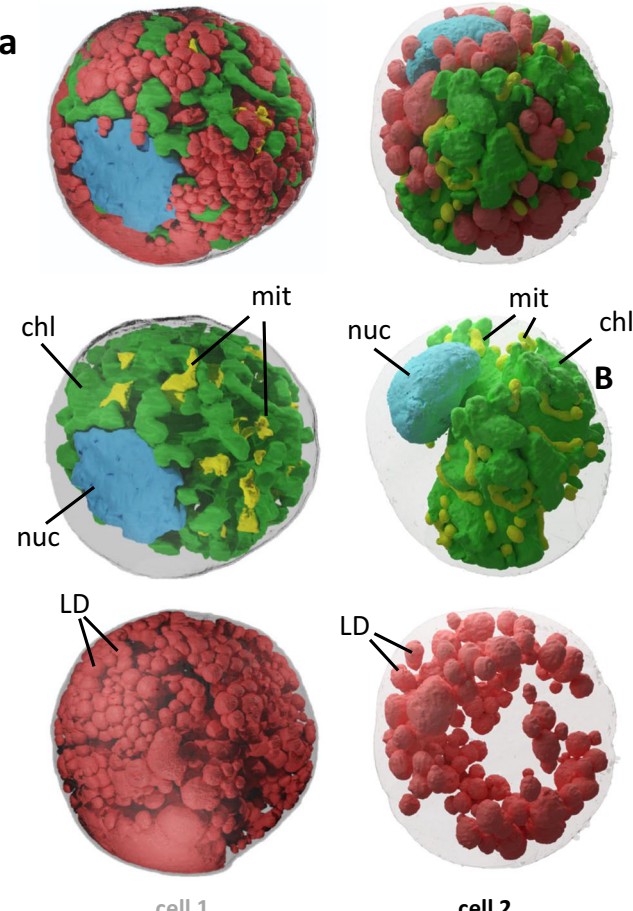

**cell 1**   **cell 2**

**b**

| | Volume | | Surface |
| --- | --- | --- | --- |
| | μm³ | % cell vol | μm² |
| cell | 535.8 - 2035.6 | 100 | 340.0 - 817.6 |
| nucleus | 12.3 - 72.5 | 2.3 - 3.6 | 49.0 - 98.1 |
| chloroplast | 152.9 - 641.7 | 28.6 - 31.5 | 740.0 - 1039.6 |
| mitochondria | 9.4 - 44.6 | 1.8 - 2.2 | 136.0 - 334.5 |
| lipid droplets | 189.0 - 270.0 | 13.3 - 35.3 | 1.127 - 1.458 |
| nuc+chl+mit | 174.6 - 758.8 | 32.6 - 37.3 | |

| | Volume ratio | Surface ratio |
| --- | --- | --- |
| chl/mit | 14.4 - 16.3 | 3.1 - 5.5 |

**Fig. 7 | 3D cell architecture of *S. nivaloides* cysts. a** Analysis of chemically- and cryo-fixed cysts. Two cells are represented, cell 1 chemically fixed and cell 2, cryo-fixed. The internal organization shows the nucleus in blue, chloroplast in green, mitochondria in yellow, and lipids droplets in red. **b** Quantitative volumetric analysis of cell compartments. Data are shown in grey for cell 1 and in black characters for cell 2. nuc nucleus, mit mitochondria, chl chloroplast, LD lipid droplet.

The nucleus is at the periphery of the cell, and its volume ranges from ~500 to ~2,000 μm³ given the level of compression by cytosolic LDs (Fig. 7a). This lateral position suggests that *S. nivaloides* nuclei are not protected from damaging UV light by astaxanthin concentrated in LDs, as suggested previously[2]. This feature is coherent with the fact that UV light may not be a critical DNA-damaging radiation in the snow habitat. On the one hand, the snow cover is known to reflect most of the incident UV radiation and to protect *Bacillus subtilis* spores used as "biological dosimeters" for exposure to harmful UV radiations[53]. On the other hand, polar regions where *S. nivaloides* is also present, have much lower exposure to UV radiance[54], unlikely to provoke any strong mutagenesis in corresponding latitudes, suggesting that nuclei protection may not be an adaptive trait in *Sanguina*.

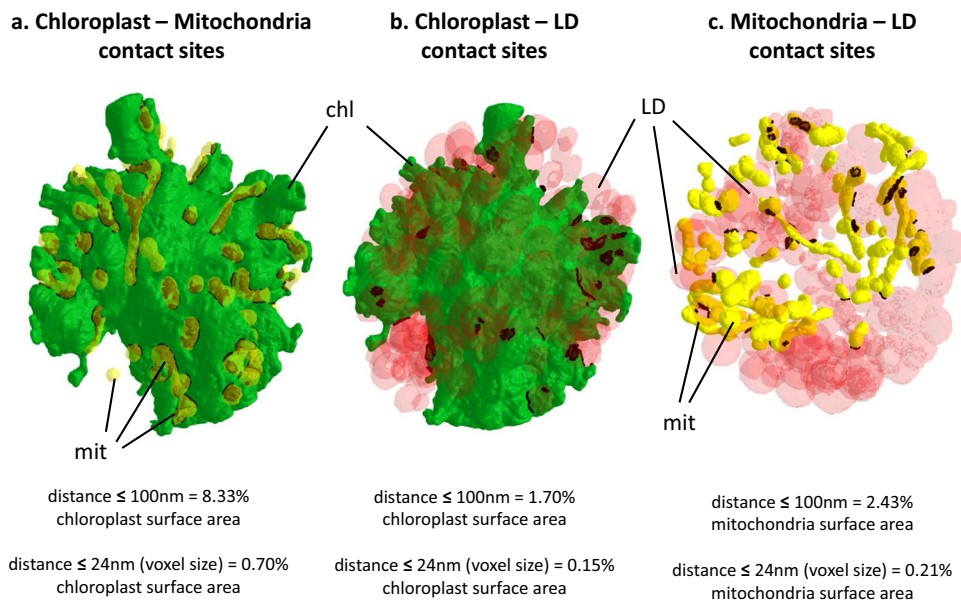

**a. Chloroplast – Mitochondria contact sites**

**b. Chloroplast – LD contact sites**

**c. Mitochondria – LD contact sites**

distance ≤ 100nm = 8.33%
chloroplast surface area

distance ≤ 100nm = 1.70%
chloroplast surface area

distance ≤ 100nm = 2.43%
mitochondria surface area

distance ≤ 24nm (voxel size) = 0.70%
chloroplast surface area

distance ≤ 24nm (voxel size) = 0.15%
chloroplast surface area

distance ≤ 24nm (voxel size) = 0.21%
mitochondria surface area

**Fig. 8 | Proximity between cellular organelles in *S. nivaloides* cysts.**
**a** Chloroplast - mitochondria. **b** Chloroplast - lipid droplets. **c** Mitochondria – lipid droplets. Distances were measured as described earlier[24]. Green, plastid surface; yellow, mitochondria surface; red, lipid droplets surface. Dark spots highlight proximity surfaces corresponding to points at a distance ≤ 100 nm between organelles. chl chloroplast, LD lipid droplet, mit mitochondria.

The plastid/mitochondria volume and surface area ratios (14.4–16.3 and 3.1–5.5, respectively) were twice that usually reported in unicellular algae[24], suggesting a very high activity in carbon assimilation (chloroplast) compared to consumption (mitochondria) (Fig. 7a). The chloroplast is central (Fig. 7a), fully immersed in a multitude of astaxanthin-filled lipid droplets, likely to protect the photosynthesis machineries from deleterious effects of high light radiance, including plastid lipid oxidation (Supplementary Table 2), consistently with previous hypotheses for the role of this carotenoid[2].

The uneven and expanded envelope membranes make the chloroplast surface extremely developed (740–1040 μm²) (Fig. 7a). This expanded surface suggests strong interactions between the chloroplast and the cytosol (via transporters) and other organelles. Indeed, > 8% of the chloroplast surface is close to mitochondria in a distance ≤ 100 nm; 0.7% of the chloroplast surface was ≤ 24 nm from mitochondria, which could reflect actual membrane contact sites. Conversely, 25.6% and 3.2% of the mitochondria surface was in the vicinity of the chloroplast, at distances ≤ 100 nm and ≤ 24 nm respectively (Fig. 8a).Thus, although chloroplast and mitochondria have little membrane contact sites, their topology within the cell suggests a functional organization.

By contrast, chloroplast contacts with LDs were far less developed (1.7% and 0.15% of the chloroplast surface was found close to LDs, with a distance ≤ 100 nm and ≤ 24 nm respectively) (Fig. 8b). This indicates that, although the plastid provides lipid and carotenoid precursors making up the core of LDs, it does not play any direct role in LD biogenesis, differing to that extent from other green algae[55].

The 3D morphometric analysis of *S. nivaloides* cysts reveals therefore the development of a hypertrophied chloroplast, suggesting an important role in carbon capture, with an expanded envelope surface, suggesting intense exchanges with the cytosol and other organelles, likely exporting precursors to load LDs with triacylglycerol and carotenoids. Proximity with mitochondria also suggests that photosynthesis and respiration may be functionally coupled in the overall carbon and energy metabolism of the cysts.

## Plastid architecture and its relation with mitochondria

*S. nivaloides* chloroplast shows specialized sub-organellar zones, separating the light-dependent reactions of photosynthesis, occurring in the thylakoids and using light energy to split water and generate ATP and NADPH, from the $CO_2$-fixing reaction[56,57]. In *S. nivaloides*, thylakoids are piled loosely by sets of about 10-15 membrane sacks (Supplementary Fig. 5a). On cyst cross sections, a piecewise segmentation of the thylakoid membranes was performed, and a comparison of thylakoid pile patterns was established for the XY orientation of the section. We could thus identify three general thylakoid pile patterns based on the organization of parallel and or branched thylakoids. (Supplementary Fig. 5a). These patterns were found repeated inside the chloroplast, with angles shifting from 0 to ± 180° (Supplementary Fig. 5a), suggesting a biogenesis process forming photosynthetic membrane units oriented with various angles. Such organization of thylakoid units is thus capable of capturing incident light coming from all possible directions. It appears as an adaptive trait to a life in the upper part of the snowpack, where light is strongly reflected and scattered by ice grains, making up an environment where cysts are literally bathed in high light coming from all directions[58]. This architecture further supports the role of astaxanthin-filled LDs as a protective peripheral screen for excessive light radiance. This also suggests that other structures may be mobilized to protect thylakoids from high light and oxidative stresses, such as plastoglobules found abundantly inside *S. nivaloides* cyst chloroplast.

As reported earlier[10,12] the chloroplast of *S. nivaloides* contains a 'layer' of plastoglobules, underneath thylakoid piles, and a central pyrenoid (Supplementary Fig. 5b). Plastoglobules form a lipoproteic compartment of still elusive role. They are considered to contribute to protective mechanisms against oxidative and high light stress, but also to support formation, remodeling, and controlled dismantling of thylakoids during developmental transitions and environmental responses[59]. Plastoglobules in *S. nivaloides* cysts may be involved in numerous, if not all, of these roles. The pyrenoid is a $CO_2$ concentration and fixation sub-compartment. In previous comparative studies of pyrenoid-containing microalgae, the pyrenoid volume represents usually 2.8-10% of the chloroplast volume[24]. Here, it occupies about 5% of the chloroplast volume (Supplementary Fig. 4b), showing that it is likely functional in enhancing the photosynthetic $CO_2$-fixing activity of the Calvin-Benson cycle enzyme Rubisco[60].

Altogether, the chloroplast architecture shows therefore a remarkable adaptation level, ensuring the capture of light scattered in

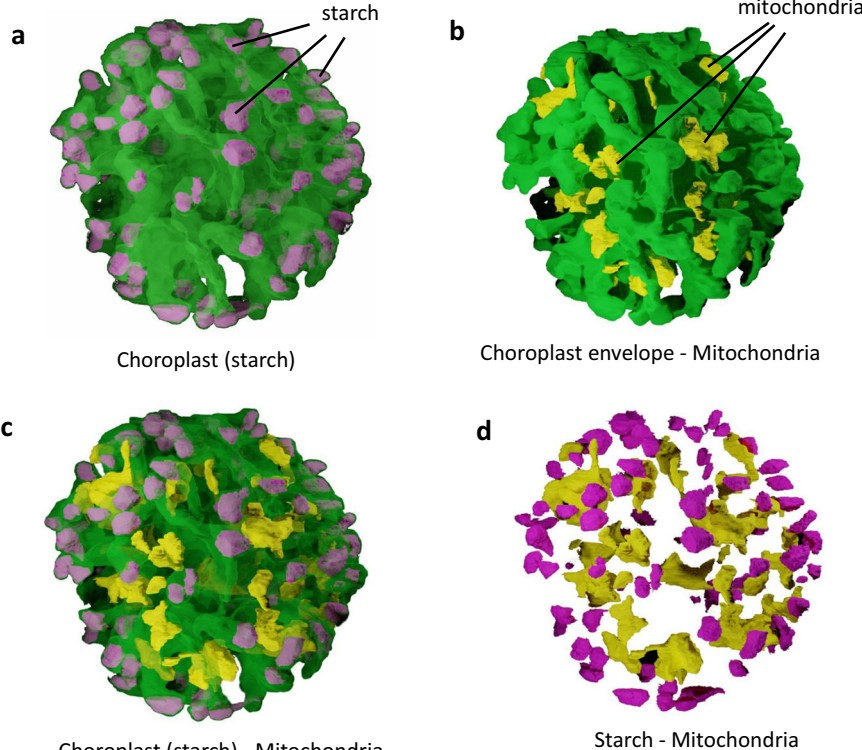

**Fig. 9 | Proximity of mitochondria to chloroplast peripheral ends filled with starch. a** Localization of starch in protuberances at the periphery of the chloroplast. **b** Mitochondria proximity to the chloroplast envelope. **c** Relative localization of chloroplast, starch, and mitochondria. **d** Mitochondria distance to starch.

Starch is visible in cells chemically fixed on the day of sampling, whereas it has been hydrolyzed in cells stored 5 days in the dark at 4 °C. The internal organization represents the chloroplast envelope, in green, starch in purple, and mitochondria, in yellow.

the snow physical environment by thylakoid units oriented in all angles, generating ATP and NADPH, H$^+$ used for the CO$_2$ fixation and reduction in the central pyrenoid. Expanded chloroplast envelope and wavy surface of the sponge-like pyrenoid (Supplementary Fig. 5b) points also to an optimized influx of CO$_2$/HCO$_3^-$. Gaps at the surface of the pyrenoid or tubules crossing its core[61] clearly visible in *S. nivaloides* (Supplementary Fig. 5b) can also enhance the inward fluxes of ATP, NADPH, H$^+$ and the exit of 3-phosphoglycerate (3PG) generated by the activity of Rubisco. 3PG is then a precursor for the biosynthesis of initial photosynthates, including starch, inside the stroma of the chloroplast, but also fatty acids exported to generate TAG in cytosolic LDs.

Starch was only observed in cyst cells fixed on the day of collection, whereas it had completely disappeared after five days of storage at 4 °C. The distribution of starch grains inside chloroplasts was striking, located exclusively at the very tip of chloroplast protrusions (Fig. 9a). Since mitochondria were at the periphery of the cell, close to the same chloroplast protrusions (Fig. 9b), the distance between mitochondria and starch (Fig. 9c, d) ranged from 25.9 to 1808.7 nm. Such proximity is likely to allow an efficient and rapid consumption of this carbon storage form.

Since after five days at 4 °C in the dark, all starch had disappeared but LDs had remained, the accumulation of starch at the tip of plastid protuberances, close to mitochondria, provides an ultrastructural organization allowing a compartmentalized consumption of carbon storage forms. Starch appears as a short-term storage form, easily accessible to the mitochondrial Krebs Cycle, whereas TAG accumulates massively in the cytosol for longer terms.

### Biogenesis of lipid droplets, with a stepwise loading of astaxanthin

Cryo-fixation of cyst cells revealed a sub-organellar organization of LDs, with the presence of numerous electron-dense spheres inside

LDs, of various numbers and size, sometimes leaking from its surface (Supplementary Movies 8 and 9; Fig. 10a). Chemical fixation did not allow the preservation of these nanostructures, LDs appearing as organelles difficult to delineate inside the cytosol, with an apparently uniform content of high electron density (Supplementary Movie 4). The electron microscopy visualization of nanostructures inside LDs is not common, but it has been reported in the algal species accumulating the highest known level of astaxanthin, i.e. *Haematococcus pluvialis*[62]. We postulate therefore that these nanostructures correspond to a liquid phase rich in carotenoids in a bulky phase of TAG.

We classified cryo-fixed LDs based on the presence and size of nanostructures corresponding to this liquid phase, with LD 'stages' numbered from (1) to (5). These stages are coherent with a stepwise loading of LDs with TAG and astaxanthin and their maturation (Fig. 10a). Stage (1) corresponds to the smallest LDs, loaded with TAG but empty of carotenoids, budding from the ER. Stage (2) is still bound to the ER but contains a multitude of tiny liquid droplets, hypothesized to contain carotenoids. These liquid nanostructures fuse together in stage (3). This stage also shows interactions with mitochondria, LD being possibly a source of TAG for fatty acid degradation in this organelle. However, mitochondria are 'better' positioned to incorporate degradation products from starch, i.e. glucose, broken down via plastid or cytosolic glycolysis. Stage (4) exhibits merged carotenoid liquid nanostructures of larger size, and stage (5) corresponds to a mature LD with a unique carotenoid liquid reservoir (Fig. 10a). Stages (4) and (5) do not interact with the ER, mitochondria or any other organelle. They appear therefore as possibly more stable cytosolic structures, able to remain for longer periods. The 'unhooking' of mature LDs from the ER or mitochondria allows their accumulation inside the cytosol, packing all possible space available, pressing the nucleus at the periphery of the cell, without any breakage thanks to the strength of the cell wall. The LD biogenesis process appears therefore

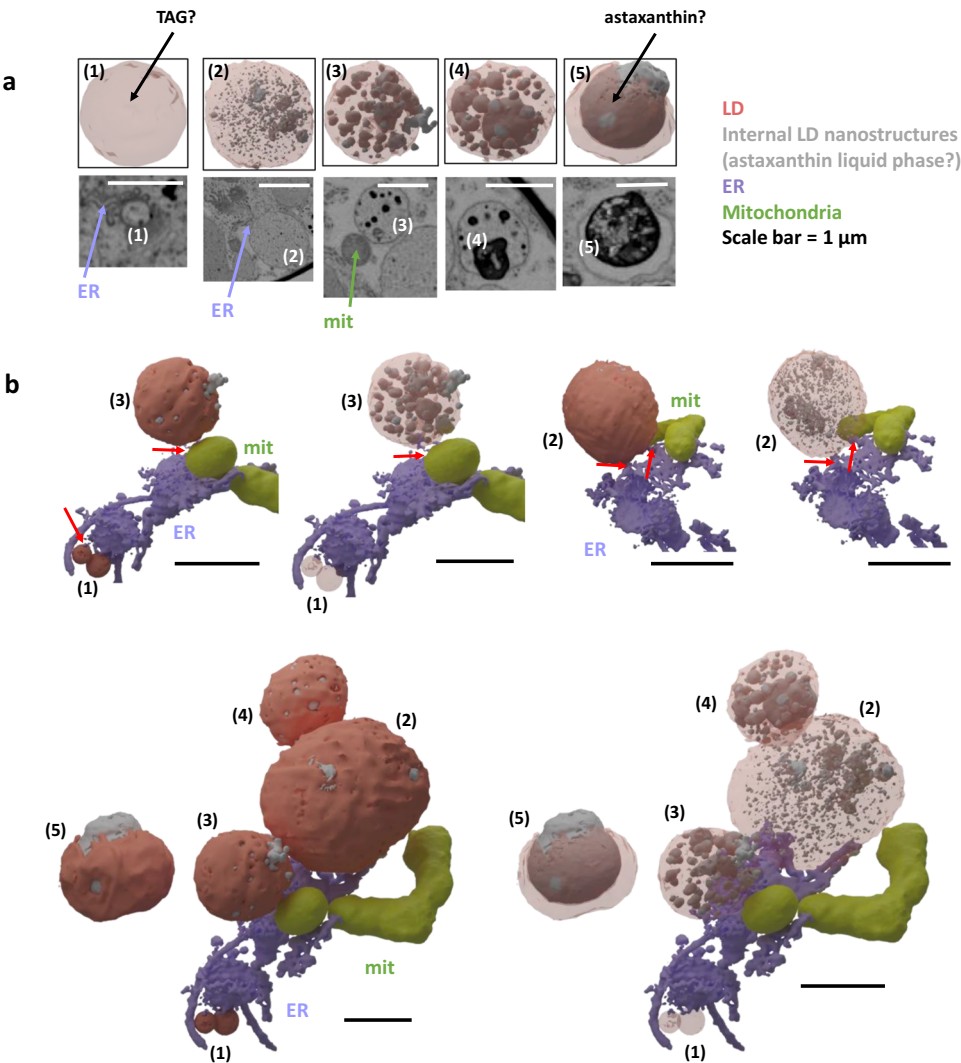

**Fig. 10 | Lipid droplet biogenesis and astaxanthin loading. a** Different stages of cytosolic lipid droplet formation from stage 1 to a mature stage 5 filled with astaxanthin. Details of astaxanthin-rich liquid subdomains are only visible in cryo-fixed cells. Stages are defined based on their size, connection with the endoplasmic reticulum, shown in purple, and mitochondria, in yellow, and density of astaxanthin domains. Mature LDs are disconnected from mitochondria, suggesting that the TAG they contain is not metabolically available for β-oxidation, and may therefore be stored for long periods. This observation has been made on 2 other cells in the FIB-SEM stack of cryo-fixed *S. nivaloides* cysts with similar result. **b** Interaction of lipid droplets with the endoplasmic reticulum and mitochondria. Mit mitochondria, ER endoplasmic reticulum, LD lipid droplet.

as a key adaptive trait of *S. nivaloides* cysts at multiple levels. On the one hand, it allows the production of astaxanthin-rich LDs acting as protective organelles against high light radiance and oxidative stress; on the other hand the same LDs accumulate TAG, a carbon-dense storage form, in organelles disconnected from the mitochondria, where fatty acids are catabolized. In cell with high LD density, only LDs at the periphery of the cell highlight contacts with mitochondria (Fig. 8C). Whereas the architecture of the cell shows that starch is an ideal carbon storage form for a rapid consumption by the mitochondrion in short-term carbon and energy metabolic requests, TAG is securely stored to allow cysts survival for months, blended with carotenoids acting as antidotes to oxidative stress.

## Conclusion

This multidisciplinary study of *S. nivaloides* in its natural context, combining optical imaging, X-ray tomography, and physicochemical characterizations of the snow, to isolated cysts, based FIB-SEM 3D-imaging and physiological and lipidomic analyzes, illuminates unique adaptive features to the long residence of this alga in snowfields. We thus determined that cysts populate the liquid water fraction at the surface of ice grains. They are photosynthetically active, can survive at least one year at +4°C, but are highly sensitive to freezing. Photosynthesis analysis also shows an adaptation to high light conditions, and stress response, consistent with the light irradiance at the surface of snowpack.

The presence of cysts influences the composition of elements concentrated during snowmelt, with compositional profiles suggesting significant efflux of K⁺ and possibly essential metals, and assimilation of phosphorus (P). Membrane and storage glycerolipid profiles confirms that cells underwent a lipid remodeling in response to a shortage of P, with an accumulation of triacylglycerol and a non-phosphorus lipid to phospholipid replacement. The three-dimensional subcellular architecture of *S. nivaloides* cysts highlights a wrinkled plasma membrane that expands the surface by a > 10% factor, likely enhancing these ionic exchanges at the cell interface with its oligotrophic environment.

The chloroplast contains thylakoids organized in piles in all orientations, capturing light scattered in all directions. The detection

of oxidized membrane lipids is possibly due to oxidative stress related to the high light conditions, and points to the role of antioxidant molecules, such as carotenoids accumulating in plastoglobules in the chloroplast stroma, and strikingly abundant in cytosolic lipid droplets. Carbon fixation occurs in a central pyrenoid and part of photosynthates are used inside the stroma to produce starch at the tip of plastid protuberances. *S. nivaloides* exhibits a unique distribution of its mitochondria at the periphery of the cell, close to these chloroplast subdomains filled with starch grains, giving insight on an optimized carbon metabolism (Supplementary Movie 10). Such peripheral localization of mitochondria was also reported in *C. reinhardtii* under low $CO_2$ concentrations[63]. Starch is thus produced as a short-term carbon storage form, easily accessible for mitochondria, and consumed within days. The stepwise biogenesis of cytosolic droplets filled with TAG and carotenoids is also unraveled and highlights that mature lipid droplets are disconnected from the ER, their biogenetic platform, but also from mitochondria, limiting fatty acid catabolism. By contrast with starch, TAG appears therefore as a carbon storage for longer periods, possibly enabling cysts to hibernate for months. The architecture of *S. nivalis* cyst is therefore also adapted to a finely tuned temporal control of carbon storage forms for shorter and longer residence in the snow environment.

This study not only points to some structural designs well adapted to a long residence in the snow, it also highlights some Achilles' heels, in particular a high sensitivity to frost. When red snowfields are subjected to an episode of very low temperatures, or when cysts on the bare ground surface after snowmelt are exposed to icy winds, the *S. nivaloides* population may die off en masse, which may be one of the mechanisms that ends bloom expansion. As an effect of climate change, the snow seasons are shortened and glaciers retreat worldwide, and the sensitivity to frost without the thermal protection of snowpack may provoke the decline and possible extinction of *S. nivaloides* in many mountain ranges. Given the magnitude of environmental changes at high elevations and in Polar Regions, it is thus essential to address whether these mechanisms are shared with other snow alga species and in more general terms, snow microorganisms. Molecular mechanisms and genomic determinants need also to be unraveled.

## Methods

### Sample collection
Most samples used in this work were collected from a representative Alpine bloom in Vallon Roche Noir (bloom 1; 45°02'55.4"N 6°23'40.8"E) at 2318 m a.s.l., on June 18, 2021, 11:00 AM, near Jardin du Lautaret laboratory facility, France. This sampling period was thus performed after intense dust deposition in February 2021, borne by winds[64]. Prior sampling, the presence of red cysts at the surface of ice grains was verified using a 100 x field digital microscope (MaxSee). Liquid water content (LWC) was based on dielectric permittivity, measured in three independent 500-mL volumes of snow, using a concentric electrode (WISe, a2photonicsensor). Red snow was collected 10 cm below the melting snow surface to avoid contamination by airborne pollen and debris, transferred into thirty-five 50-mL polycarbonate (Falcon) tubes, and carefully conserved in melting ice at 0°C. Samples were either analyzed at the Lautaret laboratory or kept at 4°C or at -20°C for further analyzes. Aliquot fractions of 100 μL were used for microscope observation (LEICA DM750) 1 hour after sampling. Other bloom samples were also collected in the area of Vallon Roche Noire / Lautaret (bloom 2, 45°05'07.4"N 6°28'15.6"E on June 20, 2019 and June 18, 2021; bloom 3, 45°03'11.1"N 6°23'17.7"E, May 11, 2022; and bloom 4, 45°03'11.0"N 6°25'41.5"E, July 4, 2022), one in Mont Brévent, France (bloom 5, 45°56'20.509"N 6°50'51.523"E at 2136 m. a.s.l.) in the same mountain range and one in Mont Olympus, Greece (bloom 6, 40°04'34.278"N 22°21'42.24"E at 2489 m. a.s.l.) and their use for specific analyzes are indicated.

### Snow X-ray tomography
Three samples that visually contained algae were collected in Col de Cerces, France, on June 20, 2019 (position of bloom 2). They were carefully transported to Grenoble in a regulated cooler at about −15 °C. In a cold room at −10°C, the samples were cut into small cylinders (2 cm in height and 2 cm in diameter) and analyzed with an X-ray tomograph (TomoCold, DeskTom 130, RX Solutions). Settings of the tomograph were $950 \times 950 \times 950$ voxel for image size, 0.01 mm resolution, 60 kV, 133 μA and 1440 projections.

### DNA extraction and polymerase chain reaction (PCR) analyzes
Algal pellets corresponding to $5.10^5$ cells were incubated in a freeze-dryer (Christ Alpha 2-4 LSC basic) for 16 hours. Frozen cells were first crushed with disposable pellet pestle (Duran Wheaton Kimble, USA). Then the algal cells were resuspended in 600 μL of cetrionium bromide (CTAB) buffer (CTAB 18.5 mM), sodium chloride (NaCl, 1.05 M), EDTA (pH 8, 15 mM), Tris hydrochloride (Tris-HCl, pH 8, 75 mM), polyvinylpyrrolidone 40 (PVP40, 0.06 mM, Sigma-Aldrich, USA) and in 200 μL of a hydrolytic enzyme cocktail adapted from[4]: 0.5 U. of cellulose Onozuka R-10 (Yakult pharmaceutical, Japan), 0.05 U. of pectolyase Y-23 (Kyowa chemical, Japan), 0.06 U. of pectinase (Sigma-Aldrich, USA) and 0.001 U. of zymolyase (Amsbio, United Kingdom). The suspension was incubated at 37°C for 3 hours and regularly mixed by inversion. Cell lysis continued by a 1-h incubation with proteinase K (20 μL, 12 U., Thermo Scientific, USA), at 55°C. After lysis, DNA extraction was initiated by the addition of 1 volume of phenol:chloroform:isoamyl alcohol (25:24:1) (Sigma-Aldrich, USA). Then vigorous mixing was carried and tubes were left at room temperature for 15 min. Next, tubes were centrifuged for 10 minutes at 13,000 g at 4 °C. The upper phase was transferred to new 1.5 mL tubes and 1 volume of chloroform (Sigma-Aldrich, USA) was added. Samples were vigorously mixed and then centrifuged for 10 minutes at 13,000 g at 4 °C. The upper phase was transferred to Phasemaker tubes (Invitrogen, USA) and 1 volume of 1-bromo-3chloropropane (BCP; Sigma-Aldrich, USA) was added. The Phasemaker tubes were centrifuged for 10 minutes at 17,000 *g* at 4°C. Again, the upper phase was transferred into 1.5 mL microtube and one volume of cold isopropanol (2-propanol; Sigma-Aldrich, USA) and one-tenth of the volume of sodium acetate (3 M, pH 5) were added. The tubes were incubated overnight at -20°C. After, a centrifugation for 30 minutes at 13,000 g at 4°C was done to pellet the DNA. The pellet was washed with ethanol 70% and the washing solution was removed after centrifugation for 10 minutes at 13,000 g at 4°C. The DNA pellet was then air dried at room temperature and suspended in 30 μL of autoclaved distilled water. DNA quantity and quality were assessed using a Nanodrop 2000 spectrophotometer (Thermo Scientific, USA) and Qubit 4 fluorometer (Invitrogen, USA) according to the manufacturer's instructions.

For rbcL amplification, the set of primer used was pair M379 5'-GGWTTYAAAGCTCTKCGTGC-3' and M1161 5'-CATGTGCCATACGT GAATAC-3' with a modification to M379 from the original sequence[5]. PCR mixture volume was 25 μL and consisted of (final concentration): Phusion HF buffer complemented with 0.5 mM dNTP, 0.4 mM of forward and reverse primers, and 1 U. of Phusion high-fidelity DNA polymerase (Thermo Scientific, USA). In addition, 1 μL of template DNA (25 ng μL − 1) was added. Negative controls were included and the PCR program was as follows: 98 °C - 5 min, 40 × (98 °C - 10 s, 61 °C - 20 s, 72 °C - 35 s), with a final extension step at 72 °C for 5 min. Agarose gel (1%) analysis was performed for verification of the PCR products. Amplicon size is approximately 800 bp. Amplicons were then sent to Macrogen for a bidirectional Sanger sequencing. The forward and reverse sequences were checked and assembled using SnapGene v5.0.8 (sapgene.com). The consensus sequence was determined to be of *Sanguina nivaloides* through Blast alignments[6].

To obtain the complete ITS1-ITS2 sequence, we used the primer pair 34F[7], 5'-GTCTCAAAGATTAAGCCATGC-3' and 26SR_chloro[8],

5′-TTGGGCTGCATTCCCAAACAAC-3′ that generate an amplicon size of approximately 3000 bp with a partial 18 S and 28 S rDNA genes. PCR mixture volume was 25 μL and consisted of (final concentration): phusion HF buffer complemented with 0.5 mM dNTP, 0.4 mM of forward and reverse primers, 1 U. of phusion high-fidelity DNA polymerase (Thermo Scientific, USA), and 1 μL of DNA template (25 ng μL⁻¹). Negative controls were included as well. The PCR program was as follow: initial denaturation at 98 °C for 5 min, followed by 8 cycles of 98 °C for 20 s, 68 °C for 45 s minus 1 °C each cycle, 72 °C for 1 min, then 27 cycles with the same steps and same duration with the target annealing temperature of 60 °C, followed by a final extension of 72 °C for 7 min[9]. A total of 5 amplifications were sent to Macrogen (France) for a Sanger bidirectional sequencing of the ITS1-ITS2 region with the pair of primer ITS1 5′-TCCGTAGGTGAACCTGCGG-3′ and ITS4 5′-TCCTCCGCTTATTGATATGC-3′. Sequences were checked and assembled as previously described.

### ITS2 secondary structure and phylogeny reconstructions

The ITS2 sequence of *Sanguina nivaloides* from Vallon Roche Noire was extracted from the ITS1-ITS4 sequence as described above using the program ITSx v1.1.3[10]. The sequence was then folded using the online tool RNAfold WebServer[11]. The centroid secondary structure[12] was visualized and exported in SVG format with the web service forna[11]. Finally, the search for compensatory base changes (CBCs) or hemi-CBCs of the ITS2 secondary structure between Vallon Roche Noire, *S. nivaloides* holotype specimen RS_0015-2010 and *S. aurantia* holotype specimen[13] RS_0017-2010 was first performed manually and then automatically using the software 4SALE[14,15].

The rbcL (ribulose-1.5-biphosphate carboxylase) sequences of 15 and 3 strains of *S. nivaloides* and *S. aurantia*, respectively, including Vallon Roche Noire unique sequence from red blooms were phylogenetically analyzed. All sequences were aligned using MUSCLE[16] via MEGAX[17] v10.2, resulting in 616 positions. Best-fit model of nucleotide substitution was computed using ModelTest-NG[18] v0.1.7. The best model was given as follows: GTR + G (0.1883); base frequencies: A 0.2971, C 0.1859, G0.1967, T 0.3203 and rate matrix A-C 0.2432, A-G 1.4545, A-U 2.6253, C-G 0.0899, C-U 5.2740, G-U 1.0000. The phylogenetic tree was inferred through Bayesian method using MrBayes[19] v3.2.7a with a burn in value of 25% and 5,000,000 generation. Also, maximum likelihood method was applied using PhyML[20] v3.3.2 with 5000 bootstraps. *Gloeocystis* sp. and two *Sphaerocystis* sp. were used as outgroup to root the tree. All trees were visualized and exported in SVG format on FigTree v1.4.4 (http://tree.bio.ed.ac.uk/software/figtree/) for representation.

### Fluorescence-based measurements and statistical analysis

Fluorescence-based photosynthetic parameters were measured with a pulse modulated amplitude fluorimeter (MAXI-IMAGING-PAM, Heinz-Waltz GmbH, Germany) as described in[65]. Prior to the onset of the measurements, cells were acclimated to darkness for 15 min. The calculations of the different photosynthetic parameter was performed based on[66] as follows: The relative photosynthetic electron transfer rate (rETR) was calculated as $(Fm′ − F)/Fm′ × I$; $F$ and $Fm′$ are the fluorescence yield in steady state light and after a saturating pulse in the actinic light, respectively; $I$ is the light irradiance in μmol photons $m^{-2}$ $s^{-1}$; NPQ was calculated as $(Fm − Fm′)/Fm′$; $Fm$ is the maximal fluorescence yield in dark-adapted cells; the effective photochemical quantum yield of photosystem II was calculated as $Y(II) = (Fm′-F)/Fm′$.

Low temperature (77 K) fluorescence emission spectroscopy was carried as follow: *S. nivaloides* cells from blooms were frozen in liquid nitrogen. Low temperature fluorescence emission spectra were recorded with the FP-6500 spectrofluorometer (Jasco, Gross-Umstadt, Germany). The obtained data were normalized to the photosystem II emission peak at 685 nm.

Statistical analysis were performed using R v4.2.1 (R Foundation for Statistical Computing, Vienna, Austria). Plots were designed using ggplot2[67] and patchwork v1.1.2 (patchwork.data-imaginist.com). Data normality was tested using Shapiro test. For Fv/Fm means ($n = 6$), a One-way Anova test was performed followed by the post-hoc test Tuckey HSD. For rETR 22 ($n = 17$) and 337 ($n = 12$) means, Kruskal-Wallis test was conducted followed by Dunn test.

### Cell chemical and cryofixation, and sample preparation for electron microscopy

Two independent fixation methods were applied to perform electron microscopy (EM) analyzes[68]. First, 1 hour after sampling, a 15-mL fraction of red snow was thawed at ambient temperature, centrifuged at 1000 g for 5 min, and pelleted cells were chemically fixed in 0.1 M phosphate buffer (PB), pH 7.4, containing 2.5% (v/v) glutaraldehyde. Samples were washed five times in 0.1 M PB, pH 7.4, and then fixed by a 1-h incubation on ice in 0.1 M PB, pH 7.4, containing 2% osmium and 1.5% potassium ferricyanide before being washed five times with 0.1 M PB, pH 7.4. Samples were then suspended in 0.1 M PB, pH 7.4, containing 0.1% tannic acid and incubated for 30 minutes in the dark at room temperature. Samples were again washed five times with 0.1 M PB, pH 7.4, dehydrated in ascending sequences of ethanol, and infiltrated with ethanol/Epon resin mixture. Finally, the samples were embedded in Epon. In the second sample preparation, five days after sampling, a second 15-mL fraction stored in the dark at 4 °C was centrifuged at 1000 g for 5 min, and pelleted cells were cryo-fixed using high-pressure freezing (EM HPM100, Leica, Germany) in which cells were subjected to a pressure of 210 MPa at −196 °C, followed by freeze substitution (EM ASF2, Leica, Germany). For the freeze substitution, a mixture 2% (w/v) osmium tetroxide and 0.5% (w/v) uranyl acetate in dried acetone was used. The freeze-substitution system was programmed as follows: 60-80 h at -90 °C, the heating rate of 2 °C h⁻¹ to -60 °C (15 h), 10-12 h at -60 °C, heating rate of 2 °C h⁻¹ to −30 °C (15 h), and 10-12 h at -30 °C, quickly heated to 0 °C for 1 h to enhance the staining efficiency of osmium tetroxide and uranyl acetate and then back at -30 °C. The cells were then washed four times in anhydrous acetone for 15 min each at -30 °C and gradually embedded in anhydrous araldite resin. A graded resin/acetone (v/v) series was used (30, 50, and 70% resin) with each step lasting 2 h at increased temperature: 30% resin/acetone bath from -30 °C to -10 °C, 50% resin/acetone bath from -10 °C to 10 °C, 70% resin/acetone bath from 10 °C to 20 °C. Samples were then placed in 100% resin for 8-10 h and in 100% resin with the accelerator benzyl dimethylamine for 8 h at room temperature. Resin polymerization finally occurred at 65 °C for 48 h.

### FIB-SEM Imaging

Focused ion beam (FIB) tomography was performed with either a Zeiss NVision 40 or a Zeiss CrossBeam 550 microscope (Zeiss, Germany), both equipped with Fibics Atlas 3D software for tomography[24]. The resin block containing the cells was fixed on a stub with silver paste and surface-abraded with a diamond knife in a microtome to obtain a perfectly flat and clean surface. The entire sample was metallized with 4 nm of platinum to avoid charging during the observations. Inside the FIB-SEM, a second platinum layer (1–2 μm) was deposited locally on the analyzed area to mitigate possible curtaining artefacts. The sample was then abraded slice by slice with the Ga⁺ ion beam (generally with a current of 700 nA at 30 kV). Each freshly exposed surface was imaged by scanning electron microscopy (SEM) at 1.5 kV and with a current of -1 nA using the in-column EsB backscatter detector. The simultaneous milling and imaging mode was used for better stability, with an hourly automatic correction of focus and astigmatism. For each slice, a thickness of 8 nm for the cryo-fixed cells and 10 nm for the chemically fixed cells was removed, and the SEM images were recorded with a pixel size of 8 nm and 10 nm respectively, providing an isotropic voxel size of $8 × 8 × 8$ nm³ and $10 × 10 × 10$ nm³. Whole volumes were imaged

with 3500 frames of 5600 × 3700 pixels for the cryo-fixed sample, and 3600 frames of 4650 × 2300 pixels for the chemically fixed sample, corresponding to acquisition times of 117 h and 112 h on the FIB-SEM respectively.

## Image segmentation and metric geometry computations

From the aligned FIB-SEM stack, the images were binned to 30% of their original size using the Fiji software (https://imagej.net/Fiji). The region of interest denoting the cell was cropped to remove unnecessary background. The segmentation and 3D reconstruction process of the images were performed using two different image analysis software (Dragonfly and 3DSlicer) to compare the results. The 3D segmentation and reconstructions of the cell chemically fixed was achieved using Dragonfly (www.theobjects.com/dragonfly) and 3D visualization made using Blender[69]. The 3D model of the cell cryo-fixed was reconstructed in 3DSlicer software[24,70] and visualized using Blender[69] to capture a 2D picture of the cell ultrastructure in three dimensions. In summary, the surface areas and volumes were calculated using discrete mesh geometry. The surface area was directly calculated from the triangles in the mesh. The volume was obtained from the signed volume of the individual tetrahedrons. In both computations, we assume that models are watertight. We estimated the proximity distance between cell compartments using the closest points between two triangular meshes. The area under the proximity distance was quantified using the minimum distance between each vertex of the A- mesh and the B- mesh. Then we get surface mapping using face data according to a distance threshold ≤100 nm chosen based on previous state-of-the-art morphometric analyzes in animal and plant cells[71,72].

## Ionomics

For each analysis, 5 samples containing 50 mL of snow were collected at the site of sampling, with a maximal distance of 3 meters between red and white snow samples. Snow was melted and filtered on a 0.2 μm nylon filter, to remove debris, dust, and living cells. Ionomic analysis was performed directly on snow water acidified with 0.65% (w/v) nitric acid and on snow water that has been concentrated 10 to 12.5 fold (freeze-drying followed by solubilization in 0.65% (w/v) nitric acid). Samples were analyzed using an iCAP RQ quadrupole mass instrument (Thermo Fisher Scientific GmbH, Germany) operating in the standard mode (for $^{11}B$, $^{23}Na$, $^{24}Mg$, $^{25}Mg$, $^{27}Al$, $^{31}P$, $^{39}K$, $^{206}Pb$, $^{208}Pb$) and the collision mode with helium as a cell gas (for $^{31}P$, $^{39}K$, $^{44}Ca$, $^{52}Cr$, $^{53}Cr$, $^{55}Mn$, $^{56}Fe$, $^{57}Fe$, $^{59}Co$, $^{63}Cu$, $^{65}Cu$, $^{64}Zn$, $^{66}Zn$, $^{69}Ga$). Quantification was done using standard curves, corrected using the internal standards $^{103}Rh$ and $^{172}Yb$, and molar concentrations were calculated using snow liquid water content of 15 mass%.

## Lipid extraction and glycerolipid class analyzes

Glycerolipids were extracted from algal snow blooms collected in Vallon Roche Noire area (Blooms 1 and 4). The profile of total fatty acids was determined on an aliquot fraction after methanolysis, based on fatty acid methyl esters identified and quantified by gas-chromatography coupled to flame ionization detection (GC-FID). Glycerolipid classes were separated by thin layer chromatography, collected, introduced by direct infusion (electrospray ionization-MS) into a trap-type mass spectrometer (AmazonXL, Bruker), and identified by comparison with standards. Glycerolipids were extracted from algal snow blooms collected at Vallon Roche Noire. First, cells from blooms 1 and 2 were harvested by centrifugation and frozen in liquid N. Once freeze-dried, the pellets were suspended in 4 mL of boiling ethanol for 5 min to prevent lipid degradation, and lipids were extracted by the addition of 2 mL of methanol and 8 mL of chloroform at room temperature, as described earlier[73]. The mixture was then saturated with argon and stirred for 1 h at room temperature. After filtration through glass wool, cell debris were rinsed with 3 mL of chloroform:methanol

(2:1, v/v), and 5 mL of 1% (w/v) NaCl was added to the filtrate to initiate the formation of two phases. The chloroform phase was dried under argon before solubilizing the lipid extract in pure chloroform. Total glycerolipids were quantified from their fatty acids: in an aliquot fraction, a known quantity of 15:0 was added and the FAs present were transformed as fatty acid methyl esters (FAMEs) by a 1-h incubation in 3 mL of 2.5% (v/v) $H_2SO_4$ in pure methanol at 100°C[74]. The reaction was stopped by the addition of 3 mL of water and 3 mL of hexane. The hexane phase was analyzed by GC-FID (Perkin-Elmer) on a BPX70 (SGE) column. FAMEs were identified by comparison of their retention times with those of standards (Sigma-Aldrich) and quantified by the surface peak method using 15:0 for calibration. Extraction and quantification were performed three times. To quantify the various classes of non-polar and polar glycerolipids, lipids were separated by thin layer chromatography (TLC) onto glass-backed silica gel plates (Merck) using two distinct resolving systems[73]. To isolate nonpolar lipids including TAG and free FA, lipids were resolved by TLC run in one dimension with hexane:diethylether:acetic acid (70:30:1, v/v). To isolate membrane glycerolipids, lipids were resolved by two-dimensional TLC. The first solvent was chloroform:methanol:water (65:25:4, v/v) and the second was chloroform:acetone: methanol:acetic acid:water (50:20:10:10:5, v/v). Lipids were then visualized under UV light, after spraying with 2% (v/v) 8-anilino-1-naphthalenesulfonic acid in methanol, and scraped off the plate. Lipids were recovered from the silica powder after the addition of 1.35 mL of chloroform:methanol (1:2, v/v) thorough mixing, the addition of 0.45 mL of chloroform and 0.8 mL of water, and collection of the chloroform phase[75]. Lipids were then dried under argon and either quantified by methanolysis and GC-FID as described above or analyzed by mass spectrometry (MS). For MS analyzes, purified lipid classes were dissolved in 10 mM ammonium acetate in pure methanol. They were introduced by direct infusion (electrospray ionization-MS) into a trap-type mass spectrometer (AmazonXL, Bruker) and identified by comparison with standards. In these conditions, the produced ions were mainly present as $H_2$, $H^+$, $NH_4^+$, or $Na^+$ adducts. Lipids were identified by $MS^2$ analysis with their precursor ion or by neutral loss analyzes[76]. The positions of FA molecular species esterified to the glycerol backbone of the various glycerolipids were determined based on $MS^2$ analyzes. Glycerol carbons were numbered following the stereospecific number (sn) nomenclature. Depending on the nature of the glycerolipid and the type of adduct, the substituents at the sn-1 (or sn-3) and sn-2 positions are differently cleaved when subjected to low-energy collision-induced dissociation. This is reflected in $MS^2$ analyzes by the preferential loss of one of the two FAs, leading to a dissymmetrical abundance of the collision fragments. The patterns of $MS^2$ fragments for all glycerolipids[41] is the following, listing lipid classes, ion analyzed and $MS^2$ fragment property, respectively: phosphatidylcholines (PC), $[M + H]^+$, $[M + H-R_2CH = C = O]^+$ >$[M + H-R_1CH = C = O]^+$; phosphatidylethanolamines (PE), $[M - H]^-$, $[R_2COO]^-$ >$[R_1COO]^-$; phosphatidylglycerol (PG), $[M - H]^-$, $[M-H-R_2COOH]^-$ >$[M-H-R_1COOH]^-$; sulfoquinovosyldiacylglycerol (SQDG), $[M - H]^-$, $[M-H-R_1COOH]^-$ > $[M-H-R_2COOH]^-$; monogalactosyldiacylglycerol (MGDG), $[M + Na]^+$, $[M +Na-R_1COO^-]^+$ > $[M+Na-R_2COO^-]^+$; digalactosyldiacylglycerols (DGDG), $[M + Na]^+$, $[M+Na-R_1COO^-]^+$ > $[M+Na -R_2COO^-]^+$; diacylglyceryltrimethylhomoserine (DGTS), $[M + H]^+$, $[M + H-R_2COOH]^+$ > $[M + H-R_1COOH]^+$; and triacylglycerol (TAG), $[M + NH_4]^+$, $[M + NH_4-R_{1/3}COO^-]^+$ > $[M + NH_4-R_2COO^-]^+$. For lipid class profiling, lipid extracts corresponding to 25 nmol of total fatty acids were dissolved in 100 μl chloroform/methanol [2/1, (v/v)] containing 125 pmol of internal standard. Lipids were then separated by HPLC using an Agilent 1200 HPLC system with a 150 mm × 3 mm × 5 μm diol column (Macherey-Nagel), at 40°C, and quantified by MS/MS on an Agilent 6460 triple quadrupole mass spectrometer equipped with a jet stream electrospray ion source. SQDG analysis was carried out in negative ion mode by scanning for precursors of m/z − 225 at a CE of −56 eV. PG, MGDG,

and DGDG measurements were performed in positive ion mode by scanning for neutral losses of 189, 179, and 341 Da at CEs of 16, 8, and 8 eV respectively. Quantification was done by multiple reaction monitoring (MRM) of all the molecules detected in the TLC-MS experiment with 100 ms dwell time. Mass spectra were processed with the Agilent MassHunter Workstation software for lipid identification and quantification. Lipid amounts were corrected for response differences between internal standards and external endogenous lipids.

### Reporting summary

Further information on research design is available in the Nature Portfolio Reporting Summary linked to this article.

## Data availability

Electron microscopy images can be accessed Electron Microscopy Public Image Archive with the following reference, EMPIAR-11694. Computation and metric geometry for 3D reconstructions were realized using Dragonfly with scripts provided at gitlab.com/clariaddy/stl_statistics and gitlab.com/clariaddy/mindist for the cryo-fixed cells. Source data are provided with this paper.

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

## Acknowledgements

The authors thank François Tuzet, Marion Reveillet, and Julien Brondex for help in the field and Giovanni Finazzi for fruitful discussions. This work was supported by the French National Research Agency (GRAL Labex ANR-10-LABEX-04, EUR CBS ANR-17-EURE-0003, CDP Glyco@Alps ANR-15-IDEX-0002, Alpalga ANR-20-CE02-0020, DIM ANR-21-CE02-0021) and the Kilian Jornet Foundation. Authors thank Parc National des Ecrins, France, and Olympus National Park, Greece, where sampling have been performed. The Lipang platform is supported by a joint Auvergne-Rhône-Alpes region / European Union FEDER program. M.D. has received funding from the European Research Council (ERC) under the European Union's Horizon 2020 research and innovation program (IVORI, grant no. 949516). D.P. has received fundings from the Human Frontiers Science Program (RGP0046/2018).

## Author contributions

J.E. and G.V. have collected most snow alga samples, performed genomic, physiological, and metabolic characterizations and

contributed to photosynthesis and lipidomic analyzes. A.S., P.S., L.L., and J.G.V. have contributed to snow algae sampling and on-site physi-cochemical analyzes. C.U., G.S.L., B.G., P.H.J., and D.F. have contributed to FIB-SEM imaging and 3D cell reconstructions. M.L., M.S., and J.J. have performed glycerolipidomic analyzes. DP has contributed to photo-synthesis analyzes. F.D. and S.R. have performed ionomic analyzes. P.H. and M.D. have performed snow analyzes based on X-Ray tomography. A.A. has contributed to field studies, and genomic and physiological analyzes. E.M. has designed the study, and contributed to field studies, genomic, lipidomic, and physiological analyzes. All authors have con-tributed to the writing of the manuscript.

## Competing interests

The authors declare no competing interests.
