## [Peer Review File · Nature Communications]

Adaptive traits of cysts of the snow alga *Sanguina nivaloides* unveiled by 3D subcellular imagingReviewer #1 (Remarks to the Author):

The authors used X-ray tomography and FIB-SEM to cut *Sanguina nivaloides* sequentially, clearly showing the three-dimensional structure of *Sanguina nivaloides* cells. Environmental samples were also analyzed by physicochemical and physiological characterization, revealing adaptive features of the algae's ability to survive in snow. This 3D model and its data will provide deep and broad inspiration for microbiologists and plant biologists. However, there are some problems with the manuscript and the following points must be revised;

1. X-ray tomography has less resolution than focused ion beam scanning electron microscopy and is it reliable for three-dimensional structural analysis of snow algae. Did the authors attempt to obtain higher resolution structures using cryo FIB-SEM?
2. Why does fig2c data fluctuate so much in each working condition? Is fig2 in situ fluorescence? Fig2g and fig2j lack labeling of light quanta.
3. While fluorescence data are useful here, the authors need to make the connection between chlorophyll fluorescence and microscopic observations, biomass-related measurements, and allometric analyses.
4. The authors should make the appropriate data to demonstrate that the cells are not connected to each other by physical connections on page 9: "The absence of apparent physical cell-to-cell contact sites suggests that biotic interactions may rely on the exchange of soluble material transferred via the plasma membranes of each partner, without any physical bridging or tethering systems."
5. Is it possible that the wrinkles in the plasma membrane of *S. nivaloides*' encapsulation could have arisen during the process of sample freezing in lieu of sample preparation, rather than being a property of the algal strain itself?
6. Can the conclusions of the manuscript about organelles such as chloroplasts, mitochondria, and lipid droplets be applied to other algae?

Reviewer #2 (Remarks to the Author):

This work provides new insights into the phylogeny of the green alga *Sanguina nivaloides*, producing astaxanthin-rich cysts, and their photosynthetic-metabolic (glycerolipidome) and especially ultrastructural features conducive to the natural habitat, emphasizing traits related to adaptation to the specialized niche. *S. nivaloides* occurs above the timberline, where the snow cover remains for months.

Because this alga is not cultivable (at least no reports on successful experiments exist), samples for most analyzes were collected from algal bloom in Vallon Roche Noir, at ~2,300 m a.s.l., in the French Alps, where several blooms occur. The cysts were isolated at the periphery of the ice, appeared photosynthetically active when kept in the lab at low positive temperature.

Cysts can also survive for months but are sensitive to freezing. Based on the snow water measurements and glycerolipidome profile, the cysts appeared to experience phosphorus limitation.

There are previous papers reporting comparative analysis (ecophysiology, ultrastructure, fatty acid composition) of the circumpolar orange snow alga *Sanguina aurantia* versus the cosmopolitan red snow alga *Sanguina nivaloides*, for example *Polar Biol.* 2021; 44(1): 105-117.

Here, the authors use state-of-the-art methods of electron microscopic analysis to establish unique features and peculiarities of the cysts in greater detail. For example, the authors have produced for the first-time stanning videos and micrographs that capture fine details of cellular ultrastructure, 3D cell architecture, reveal the potential astaxanthin-rich zones within lipid droplets, chloroplast topography and proximity between cellular organelles, suggesting beta-oxidation of fatty acids from TAG as a major long term energy resource for cysts survival.

Lipidome analysis revealed oxidized MGDG species, indicative of oxidative stress. The comparison to some other chlorophytes reveals some species that have not been previously reported. Also, this paragraph can be smoothed, the major species were still shared.

Although some of the findings and statements would require experimental confirmation, this is

extremely difficult or impossible with *S. nivaloides* with has no records of successful cultivation

Point-by-point response to reviewers

Reviewer #1:

Comment 1. X-ray tomography has less resolution than focused ion beam scanning electron microscopy and is it reliable for three-dimensional structural analysis of snow algae. Did the authors attempt to obtain higher resolution structures using cryo FIB-SEM?

Response 1: X-ray tomography and FIB-SEM are distinct methods, each with its unique technical constraints and intrinsic resolutions. On one hand, we employed low-energy X-ray tomography to capture the overall 3D structure of freshly collected red snow, achieving a resolution of 10 μm . Preserving the snow structure as closely as possible to its native form is a challenge, necessitating a swift transfer of the sample from the collection site to the X-ray source. While it is feasible to reach a resolution of 5 μm , this would not yield significantly more information, given that algal cells typically range from 10 to 30 μm in diameter. This method is detailed in reference 22 (Hagenmuller P, Matzl M, Chambon G, Schneebeli M. Sensitivity of snow density and specific surface area measured by microtomography to different image processing algorithms. *Cryosphere* **10**, 1039-1054 (2016)). In contrast, FIB-SEM is an electron microscopy method capable of achieving resolutions in the nanometer (nm) range. In our study, we utilized resolutions of 8 nm for cryo-fixed cells and 10 nm for chemically fixed cells. Overall, X-ray tomography is not suitable for conducting three-dimensional structural analysis of snow algae and this is why we limited the use of X-ray tomography to examining red snow samples. Eventually, since this paper was focused on cell architecture and membrane organelle morphometry, we did not address details at the molecular level and as a result Cryo EM (Cryo FIB-SEB) was not used.

Comment 2. Why does fig2c data fluctuate so much in each working condition? Is fig2 in situ fluorescence? Fig2g and fig2j lack labeling of light quanta.

Response 2. Data of Figure 2 were collected using MAXI-IMAGING-PAM Chlorophyll Fluorometer (Walz, Germany) and therefore represent measurements of in vivo photosynthesis. The measurements were made ex situ, in the laboratory. The reason for the fluctuation in the data in Fig. 2c is likely the weak fluorescence signal of the sample as can be seen when we plot the raw fluorescence data to compare fresh (fig2a) and frozen (fig2c) in the figure shown below (left panel). Weak fluorescent signals would lead to low signal to noise ratio and would explain the bumpier data set of fig2c (better seen in the right graph of the figure shown below). At the same time F_v/F_m of the frozen sample was low which could indicate a dissociation of light-harvesting chlorophyll a/b protein complexes from the reaction center complex of photosystem II (PSII) or photodamage. These experiments, although not ideal, have been performed, provided key information on photosynthesis function and were presented in this paper, because cells are not cultivable.

Comment 3. While fluorescence data are useful here, the authors need to make the connection between chlorophyll fluorescence and microscopic observations, biomass-related measurements, and allometric analyses.

Response 3. The measurements were of a qualitative nature and aimed to investigate whether the collected/preserved algal cells were photosynthetically active, which was found to be the case. Direct comparisons between samples should be made with caution, mainly because the cell samples may have differed greatly in terms of their physiology, given that they are not cultivable. Therefore, the fluorescence data presented here should not be interpreted as absolute measures of pigments, including chlorophylls, carotenoids, or other potential fluorescent compounds within the cell wall. These data may also not be suitable for compartment detection in cell imaging, since fluorescence was not measured after a simple excitation by a wavelength, but after a precise sequence described in the text, allowing a characterization of photosystems' function. Eventually, it is important to note that these fluorescence data cannot be reliably correlated with any measurements related to biomass.

Comment 4. The authors should make the appropriate data to demonstrate that the cells are not connected to each other by physical connections on page 9: "The absence of apparent physical cell-to-cell contact sites suggests that biotic interactions may rely on the exchange of soluble material transferred via the plasma membranes of each partner, without any physical bridging or tethering systems."

Response 4. As stated in the sentence quoted by reviewer #1, we did not claim that this observation was a demonstration per se. We focused on membrane contact sites (and not cell wall interactions), and we understood that our sentence was misleading. We rephrased this part as follows: "Although bacteria were at the vicinity of *S. nivaloides* cell surface (Fig. 3d, blue arrows), suggesting that the respective cell walls may interact physically, we did not notice any direct contact site between algal and bacterial cell membranes. The absence of apparent physical cell-to-cell membrane contact sites suggests that biotic interactions may rely on the exchange of soluble material transferred via the plasma membranes of each partners, without any physical bridging or tethering systems."

Comment 5. Is it possible that the wrinkles in the plasma membrane of *S. nivaloides*' encapsulation could have arisen during the process of sample freezing in lieu of sample preparation, rather than being a property of the algal strain itself?

Response 5. If sample preparation had altered the plasma membrane by inducing wrinkles, we would expect to observe distortions and 'gaps' at the outer interface with the rigid cell wall (which is

unlikely to undergo any plastic changes parallel to the plasma membrane) and at the inner interface with the cytoplasm, where lipid droplets are densely packed. Furthermore, this distinctive feature is not present in any other cell membrane, and to our knowledge, there are no reports of such observations in other cell systems analyzed to date after cryofixation. The freezing method is known for its rapidity and its ability to better preserve cellular structures compared to chemical fixation. If the wrinkles on the plasma membrane were a result of the fixation process, it would raise questions about how the smoother plasma membrane could fit within the cell wall, given the dimensions that would be required to accommodate such an expanded structure. Overall, we find no compelling argument to suggest that the observed structure of the plasma membrane is not genuine.

Comment 6. Can the conclusions of the manuscript about organelles such as chloroplasts, mitochondria, and lipid droplets be applied to other algae?

Answer 6. The conclusions drawn here may potentially apply to adaptive traits observed in other species inhabiting snow or similar environments that share common characteristics such as low temperature, intense scattered light in multiple directions, and oligotrophy. Further research is necessary to assess the validity of these findings in broader ecological contexts

Reviewer #2 (Remarks to the Author):

Comments from Reviewer #2 are overall positive.

Reviewer #2 mentions that “There are previous papers reporting comparative analysis (ecophysiology, ultrastructure, fatty acid composition) of the circumpolar orange snow alga *Sanguina aurantia* versus the cosmopolitan red snow alga *Sanguina nivaloides*, for example *Polar Biol.* 2021; 44(1): 105-117.”

Answer: we have included this reference (reference 12 - Prochazkova L, Remias D, Holzinger A, Rezanka T, Nedbalova L. Ecophysiological and ultrastructural characterisation of the circumpolar orange snow alga *Sanguina aurantia* compared to the cosmopolitan red snow alga *Sanguina nivaloides* (Chlorophyta). *Polar Biol* 44, 105-117 (2021).) and cited this work accordingly in our initially submitted paper.

Later, Reviewer mentions that “Lipidome analysis revealed oxidized MGDG species, indicative of oxidative stress. The comparison to some other chlorophytes reveals some species that have not been previously reported. Also, this paragraph can be smoothed, the major species were still shared.”

Answer. Although previous studies, such as that of Prochazkova et al (12), have referred to the term 'lipidomics', they have typically focused on the analysis of fatty acids (FAs) alone. It is important to note that FAs can be components of various acyl-lipids, most notably membrane glycerolipids, which usually feature two FAs, though some lyso-glycerolipid classes contain only one FA and some specific acylated membrane glycerolipids accommodate a third FA grafted to their polar heads. Additionally, triacylglycerols (TAGs) comprise three FAs. Furthermore, acylation can occur in molecules beyond glycerolipids, such as carotenoids, as seen in astaxanthin fatty acid esters. As such, comparing FA profiles alone may provide limited information. For instance, one sample might be rich in TAGs, with the observed FAs primarily representing storage lipids, while another sample might have fewer oil droplets, causing the FAs to reflect the composition of membrane lipids. In reference 12, the

comparison of FAs from *S. nivaloides*- and *S. aurantia*-rich samples cannot be interpreted, without knowing the glycerolipid classes harbouring these FA. In our laboratory, specializing in lipids, we believe that the most comprehensive approach involves precise profiling of all glycerolipid classes, a task undertaken for the first time in our study. This comprehensive analysis allows for a more refined examination of lipid composition. Importantly, in our work, we did not observe oxidized forms in all lipid classes. We believe therefore it is crucial to report these detailed analyses, as an important reference for future research involving lipidomic analyses of other species.

Reviewer #1 (Remarks to the Author):

The authors have addressed my problems seriously. And the revised version is more comprehensive and it takes into account the reviewers suggestions. Therefore, I recommend its acceptance in the revised form.